# EFFICIENT REINFORCEMENT LEARNING WITH LARGE LANGUAGE MODEL PRIORS

**Xue Yan** [1 2] **, Yan Song** [3]**, Xidong Feng** [3]**, Mengyue Yang** [4]
**Haifeng Zhang**[† 1 2]**, Haitham Bou Ammar** [3 5]**, Jun Wang**[† 3]

[1]Institute of Automation, Chinese Academy of Science, Beijing, China
[2]School of Artificial Intelligence, University of Chinese Academy of Sciences, China
[3]AI Centre, Department of Computer Science, University College London, London, UK
[4]University of Bristol, UK [5]Huawei Technologies, London, UK

## ABSTRACT

In sequential decision-making tasks, methods like reinforcement learning (RL) and heuristic search have made notable advances in specific cases. However, they often require extensive exploration and face challenges in generalizing across diverse environments due to their limited grasp of the underlying decision dynamics. In contrast, large language models (LLMs) have recently emerged as powerful general-purpose tools, due to their capacity to maintain vast amounts of domain-specific knowledge. To harness this rich prior knowledge for efficiently solving complex sequential decision-making tasks, we propose treating LLMs as prior action distributions and integrating them into RL frameworks through Bayesian inference methods, making use of variational inference and direct posterior sampling. The proposed approaches facilitate the seamless incorporation of fixed LLM priors into both policy-based and value-based RL frameworks. Our experiments show that incorporating LLM-based action priors significantly reduces exploration and optimization complexity, substantially improving sample efficiency compared to traditional RL techniques, e.g., using LLM priors decreases the number of required samples by over 90% in offline learning scenarios.

## 1 INTRODUCTION

Many real-world tasks, including robotics (Polydoros & Nalpantidis, 2017; Brunke et al., 2022; Rana et al., 2023), autonomous driving (Naranjo et al., 2005; Song et al., 2022), human-AI dialogue (Li et al., 2019; McTear, 2022), and human-AI gaming (Granter et al., 2017; Yan et al., 2024), involve complex sequential decision-making (SDM) challenges. Traditional approaches to SDM, such as optimal control (Garcia et al., 1989), heuristic search (Świechowski et al., 2023) and reinforcement learning (RL) (Mnih, 2013), have seen substantial success. Notably, AlphaGo (Silver et al., 2016) and AlphaStar (Vinyals et al., 2019), both based on deep reinforcement learning (DRL), have achieved human-level proficiency in the games of Go and StarCraft II, respectively. However, these methods still suffer from high computational complexity, along with poor generalizability and limited applicability across diverse domains (Dulac-Arnold et al., 2015; Cobbe et al., 2019).

Recently, Large Language Models (LLMs) have emerged as effective tools for tackling diverse general-purpose tasks, such as in dialogue systems (Brooks et al., 2023), decision-making (Zhao et al., 2024a), and mathematical reasoning (Imani et al., 2023). Their impressive performance across various domains is largely attributed to the vast amounts of human knowledge compressed during the pre-training phase on extensive corpora (Tucker & Tuckute, 2023). Inspired by how humans make decisions using existing knowledge while learning from new tasks, LLM-based agents are emerging as a promising approach for solving SDM tasks. For instance, researchers leverage human-crafted

---

[†]Correspondence to : ⟨haifeng.zhang@ia.ac.cn ⟩, ⟨jun.wang@cs.ucl.ac.uk⟩

prompts to guide LLMs in making decisions directly (Zhang et al., 2023; Wang et al., 2023; Ma et al., 2023). While this approach relies heavily on the quality of the prompts and the LLMs' inherent capabilities, another line of research aims to fine-tune LLMs with RL algorithms (RLFT) (Carta et al., 2023; Christianos et al., 2023; Tan et al., 2024; Zhou et al., 2024) to enhance their decision-making precision. A more detailed exploration of related work can be found in Section 5. However, these approaches are resource-intensive, as they require meticulously crafted human prompts and an expensive fine-tuning process for LLMs for each specific task.

Inspired by these, this work stands in the position of combining the advantages of leveraging rich domain knowledge with the strengths of reinforcement learning (RL), but in a different perspective to use fixed LLMs to enhance traditional RL. The key insight is that while it may be challenging for an LLM to generate optimal plans at every state with limited experience, it can still offer suboptimal action proposals or significantly reduce the size of the action space to a manageable scope, thereby alleviating computational burdens. Building on this idea, we treat the LLM as an action prior distribution and incorporate it into Markov Decision Processes (MDPs) solving from a Bayesian inference perspective (Hu et al.). The primary objective is to approximate the posterior action distribution that aligns with task goals through probabilistic inference approaches such as variational inference or direct posterior sampling. Building on these approaches, we derive the incorporation of LLMs into both policy-based and value-based RL frameworks in a simple yet logically sound manner. We conduct experiments on major benchmarks like ALFWorld and Overcooked, demonstrating that the sample efficiency of traditional RL algorithms, such as value-based DQN and policy-based PPO, can largely benefit from integrating LLM action priors.

In summary, our main contributions are three-fold:

1. We present a unified framework for integrating Large Language Models (LLMs) as probabilistic priors into Markov decision-making frameworks. **(Section 2)**

2. We practically implement the framework by leveraging LLMs as the refined action sampler for value-based online RL **(Section 3.1)**, offline RL **(Section 3.2)** or behavior regularizer for policy-based RL **(Section 3.3)**.

3. Extensive experiments on ALFWorld and Overcooked demonstrate that our new framework can significantly boost sample efficiency compared with both pure RL and pure LLM baselines, and also bring in more robust and generalizable value function. **(Section 4)** Source code is available at `https://github.com/yanxue7/RL-LLM-Prior`.

## 2 FORMULATION

We seek to utilize LLMs to solve complex SDM tasks from a Bayesian inference perspective. Although LLM struggles to directly generate optimal plans in sequential decision-making tasks, it can still provide valuable suggestions for possible sub-optimal actions thanks to its rich prior knowledge Hao et al.; Zhang et al. (2024b). Building on this insight, we treat the LLM as a reliable prior distribution over valid actions and analyze its role in solving MDPs from a Bayesian inference perspective, using traditional probabilistic tools such as variational inference and probabilistic sampling.

**Markov Decision Process** A Markov Decision Process (MDP) can be described as a tuple $\langle \mathcal{S}, \mathcal{A}, P, r, \gamma \rangle$. $\mathcal{S}$ is the state space, and $\mathcal{A}$ is the finite action space. We consider the textual action space and state space here. Each state $s$ and each action $a$ is a sentence $s, a \in \mathcal{V}^\infty$, where $\mathcal{V}$ is the vocabulary set. $P : \mathcal{S} \times \mathcal{A} \to \Delta(\mathcal{S})$ is the transition kernel, mapping the current state $s_t$ to the next state $s_{t+1}$ following the action $a_t$. $r : \mathcal{S} \times \mathcal{A} \to \mathbb{R}$ is the reward function. $\gamma$ is the discount factor.

**LLM as an action prior** It is challenging to directly solve textual decision-making tasks due to the lack of task-specific experience, yet powerful LLMs demonstrate the ability to generate reasonable action proposals Hao et al.; Yao et al. (2024). To maximize the potential of LLMs in decision-making, we treat the LLM as an action sampler, denoted as $p_{\text{LLM}}(a|s_t)$. Regarding the implementation details of sampling from LLM priors, it is challenging for an LLM to output an executable action directly. Therefore, we prompt the LLM with the textual state and admissible

actions, then get a free-form output from the LLM (e.g., a 7B LLM). This output is subsequently mapped to an executable action through a simple rule-based projection, formally described as follows: $p_{\text{LLM}}(a|s_t) = \sum_o p(a|o)\text{LLM}(o|s_t)$, where $p(a|o)$ is a projection from the LLM output $o$ to action $a$. More detailed explanations of the LLM prior setting can be found in Appendix 9.2.

## 2.1 VARIATIONAL INFERENCE FOR MARKOV DECISION PROCESS

Similar to the *Control as Inference* framework proposed by (Levine, 2018), we introduce a binary optimality variable $\mathcal{O}$ to indicate the quality of a trajectory $\tau = \{s_0, a_0, r_0, s_1, ..., s_n\}$, where $\mathcal{O} = 1$ represents an optimal/successful trajectory, and $\mathcal{O} = 0$ otherwise. Hence, the likelihood, positively related to the reward, can be written as $p(\mathcal{O} = 1|\tau) \propto \exp\left(\sum_t \gamma^t r_t/\alpha\right)$, with $\alpha$ as the temperature parameter. Our goal is to maximize the optimal marginal distribution. To do that, we first apply variational inference:

$$p(\mathcal{O} = 1) \geq \mathbb{E}_{q(\tau)}\left[\log p(\mathcal{O} = 1|\tau)\right] - \text{KL}\left[q(\tau)\|p(\tau)\right] \tag{1}$$

We define the prior $p(\tau)$ as the trajectory distribution, which follows the factorization: $p(\tau) = p(s_0)\prod_t p(a_t|s_t)P(s_{t+1}|s_t, a_t)$. Here, we specify $p_{\text{LLM}}(a|s)$ as the action prior $p(a|s)$. The variational distribution $q(\tau)$ follows a similar factorization: $q(\tau) = p(s_0)\prod_t q(a_t|s_t)P(s_{t+1}|s_t, a_t)$. Substituting these factorizations back into Eq. 1, we obtain the step-wise objective as:

$$\arg\max_\theta \sum_t \mathbb{E}_{\pi_\theta}\left[\gamma^t r_t\right] - \alpha KL\left[\pi_\theta(a_t|s_t)\|p_{LLM}(a_t|s_t)\right], \tag{2}$$

and we learn a language model policy $\pi_\theta$ as the variational distribution $q(a_t|s_t)$, with parameter $\theta$. More detailed derivation can be found in Appendix 8.1.

However, fine-tuning language models in policy-based methods can be challenging in practice. To address this, we propose an alternative approach: **direct posterior inference**, which allows us to sample directly from the optimal posterior distribution $p(\tau|\mathcal{O} = 1)$ for sequential decision-making.

## 2.2 DIRECT POSTERIOR INFERENCE FOR MARKOV DECISION PROCESS

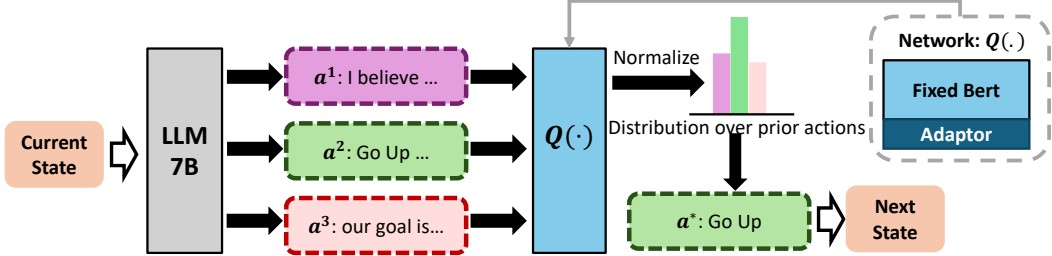

Figure 1: An illustration of the process of approximate sampling from the intractable posterior: $q(a|s, \mathcal{O} = 1) \propto p(\mathcal{O} = 1|a, s_t)p_{\text{LLM}}(a|s_t)$, implemented by reweighting the action prior proposals according to $Q$-values, which act as likelihood estimates. In our experiments, the $Q$ function adopts BERT to encode textual state-action pairs and output a scalar value through an adaptor network.

Since the optimal posterior distribution over trajectory $\tau$ can be factorized into step-wise posterior described as: $p(\tau|\mathcal{O} = 1) \propto p(s_0)\prod_t p(a_t|s_t, \mathcal{O} = 1)P(s_t|s_{t-1}, a_{t-1})$, the learning procedure can be divided into two stages, modeling and inference of step-wise posterior $p(a|s_t, \mathcal{O} = 1)$. The modeling stage will point out and decompose the desirable distribution into more solvable probabilistic terms. The inference stage will give the desirable outputs using sampling-based techniques (Korbak et al., 2022).

**Posterior Modeling** We first use Bayes' rule to translate the intricate posterior into accessible terms:

$$p(a|s_t, \mathcal{O} = 1) = \frac{p(\mathcal{O} = 1|a, s_t)p(a|s_t)}{p(\mathcal{O} = 1|s_t)} \propto p(\mathcal{O} = 1|a, s_t)p(a|s_t), \tag{3}$$

which is a combination of the likelihood $p(\mathcal{O} = 1|a, s_t)$ and the prior $p(a|s_t)$. Here, we use the LLM as the prior action distribution $p(a|s_t) \leftarrow p_{\text{LLM}}(a|s_t)$. Recall that the optimality likelihood in MDPs

is given by: $p(\mathcal{O} = 1|a, s_t) \propto \exp\left(\sum_{i=t} \gamma^{i-t} r_i\right)$, which corresponds to whether the goal state is achieved, similar to the concept of the Q function $Q(s_t, a)$ in RL. In practice, methods such as MCTS or TD-learning can estimate the optimality likelihood $p(\mathcal{O} = 1|s_t, a)$.

**Action Inference**  Having described the posterior as a product between LLM prior $p_{LLM}(a|s_t)$ and estimate Q function $Q^\theta(s_t, a)$, we can implement the inference from the posterior distribution $p(a|s_t, \mathcal{O} = 1)$ by following these steps:

- Sample actions based on the prior policy $p_{\text{LLM}}$: Obtain $k$ action candidates for the current state $s_t$, denoted as: $\mathcal{C}^k(s_t) = \{a_1, a_2, \cdots, a_k\}$, where $a_i \sim p_{\text{LLM}}(a|s_t)$.
- Choose the next action from the candidates based on the estimated Q-values: Select an action $a^*$ according to the softmax probability distribution over the estimated Q-values. Specifically, $a^* \sim \text{softmax}(Q^\theta(s_t, a_1)/\alpha, Q^\theta(s_t, a_2)/\alpha, \ldots, Q^\theta(s_t, a_k)/\alpha)$, where $\alpha$ is a hyper-parameter.

This sampling strategy is illustrated in Figure 1, where the action candidates are reweighted based on the Q-values. Below, we establish the connection between variational inference and direct posterior inference.

**Proposition 1.** *The above action inference strategy—selecting an action from the action priors reweighted based on Q-values—can be described as following a distribution q. As $k \to \infty$, we have:*

$$\lim_{k\to\infty} q(a|s_t) = p_{LLM}(a|s_t)\exp(Q^\theta(s,a)/\alpha)/\mathbb{E}_{a_j \sim p_{LLM}(\cdot|s)} \exp(Q^\theta(s,a_j)/\alpha) \tag{4}$$

*The limiting policy corresponds to the policy that optimizes the Q-values with a KL regularizer:*

$$\lim_{k\to\infty} q(\cdot|s_t) = \arg\max_\pi \mathbb{E}_{\pi(a|s_t)}[Q^\theta(s_t,a)] - \alpha KL\left(\pi(\cdot|s_t)\|p_{LLM}(\cdot|s_t)\right) \tag{5}$$

*Then, the posterior sampling strategy is highly related to the solution of variational inference as shown in Eq. 2. Proof. Please see the appendix 8.2.*

## 3 PRACTICAL IMPLEMENTATION

In this section, we introduce the practical implementation of posterior approximation within robust RL frameworks. We describe a policy-based RL method with an LLM prior to realize the variational inference problem. In the meantime, as discussed earlier, optimality likelihood estimation is critical for direct posterior inference and is positively associated with Q-values in the MDP setting. Thereby, we consider Q-value estimation with an LLM prior for value-based RL in both online and offline settings. Finally, we propose three simple yet efficient RL algorithms: one policy-based RL algorithm and two Q-learning variants.

We will first introduce the two Q-learning variants, which perform exploration and optimization within a narrowed yet reliable LLM prior action space, eliminating the need for fine-tuning the language model-based policy. Then, we will present a policy-based PPO variant that incorporates a KL loss with respect to the LLM prior. Detailed descriptions of classic Q-learning algorithms, DQN Mnih (2013) and CQL Kumar et al. (2020), can be found in the Appendix 7.

### 3.1 ONLINE VALUE-BASED RL WITH LLM PRIOR

The Q-value estimation is based on DQN (Mnih, 2013), a widely used value-based RL method, under the online setting. Unlike traditional DQN, which performs exploration and Bellman updates over the full action space $\mathcal{A}$, we propose a variant with an LLM prior, referred to as DQN-Prior. This variant limits these processes to the prior action space $\mathcal{C}^k \subseteq \mathcal{A}$, which is sampled from $p_{\text{LLM}}$. This prior action space represents a reduced and more rational subspace where clearly irrelevant actions may be filtered out in advance. Specifically, DQN-Prior alternates between the following steps:

**Exploration**  Roll out new episodes using the posterior inference strategy as described in Sec. 2.2, specifically:

- For a state $s_t$, sample the action prior space $\mathcal{C}^k(s_t) = \{a_1, a_2, \cdots, a_k\}, a_i \sim p_{\text{LLM}}(a|s_t)$.

- Then sample the action $a_t \sim \text{softmax}\left(Q^\theta(s_t, a_1)/\alpha, \cdots, Q^\theta(s_t, a_k)/\alpha\right)$.
- Apply the action $a_t$ to the environment, and obtain the next state $s_{t+1} \sim P(\cdot|s_t, a_t)$.
- Add the tuple to the replay buffer: $\mathcal{D} = \mathcal{D} \bigcup (s_t, a_t, s_{t+1})$.

**Q-function Update** After expanding the replay buffer, we use TD-learning to update the Q-network. The loss function for the $i$-th iteration is given as:

$$\mathcal{L}_i(\theta) = \mathbb{E}_{(s,a,s')\sim\mathcal{D}} \left[ (Q^{\theta_i}(s,a) - y_i)^2 \right], \tag{6}$$

where $y_i = \mathcal{B}^* Q^{\theta_{i-1}}(s,a) = r(s,a) + \gamma \max_{a' \in \mathcal{C}^k(s')} Q^{\theta_{i-1}}(s', a')$. Compared to traditional DQN, the DQN-Prior performs exploration and applies the Bellman optimal operator within the LLM prior action space, as highlighted in red in Eq. 6.

For most experiments, we use Qwen-1.5 7B Bai et al. (2023) to generate $k = 5$ prior actions. To represent the Q-network, Bert (Devlin et al., 2019) is used to encode state-action pairs and remains frozen during optimization. Then, we introduce an adapter (a 3-layer MLP) on top of Bert's embeddings, as illustrated in Figure 1, and optimize the adapter's weights by minimizing the temporal difference loss. Therefore, we do not propagate any gradients through the Bert to avoid expensive computational overhead.

## 3.2 OFFLINE VALUE-BASED RL WITH LLM PRIOR

This study also explores the feasibility of using gradient-free LLM priors for successful SDM based solely on offline datasets. (Snell et al., 2023) have explored the use of offline RL for language generation tasks based on the SFT LLM from offline datasets. This approach assumes the entire token vocabulary as the action space, where each action corresponds to a single token, resulting in a vast action space with tens of thousands of possible actions. In our work, we consider a discrete action space where each action is a short phrase consisting of several tokens. For the SDM tasks we address, the action space is significantly smaller, with fewer than a hundred possible actions. Additionally, by leveraging an action prior, we can first obtain a tidy sub-action space $\mathcal{C}^k$, thereby narrowing the optimization space.

We apply this action prior within the CQL framework, a widely used offline RL approach, and refer to this variant as CQL-Prior. The loss function for the proposed CQL-Prior is given as:

$$\mathcal{L}_i(\theta) = \beta \mathbb{E}_{(s,a)\sim\mathcal{D}} \left[ \log \sum_{a' \in \mathcal{C}^k(s)} \exp(Q^{\theta_i}(s,a')) - Q^{\theta_i}(s,a) \right] + \frac{1}{2} \mathbb{E}_{(s,a,s')\sim\mathcal{D}} \left[ (Q^{\theta_i}(s,a) - y_i)^2 \right], \tag{7}$$

where $y_i = r(s,a) + \gamma \max_{a'\sim\mathcal{C}^k(s')} Q^{\theta_{i-1}}(s', a')$ and $\beta$ is a hyper-parameter. The main difference from traditional CQL is that it restricts the overestimation of Q-values to the tidy action prior space and applies the Bellman optimal operator within this space, as highlighted in red.

## 3.3 POLICY-BASED RL WITH LLM PRIOR

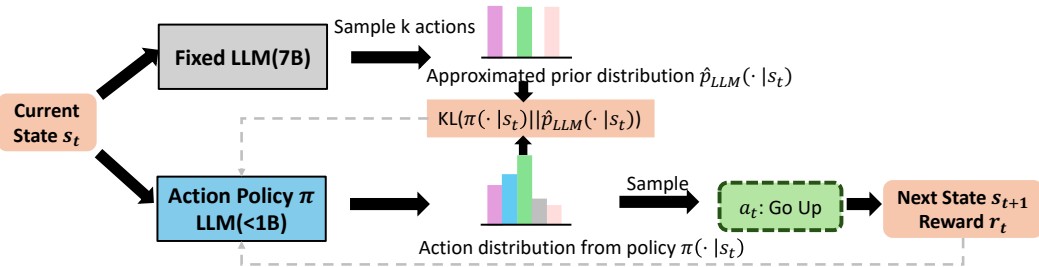

Figure 2: An illustration of the training process of GFlan-Prior, a policy-based RL algorithm with an LLM prior. A fixed LLM serves as a suboptimal prior, while the action policy learns to approximate the posterior action distribution, guided by both environmental rewards and the LLM prior.

We implement variational inference for MDPs with LLMs as action priors using a policy-based RL framework. Specifically, we build upon GFlan (Carta et al., 2023), let the Flan-T5 small model (Rae et al., 2021) (with less than $1B$ parameters) as the action policy for leveraging pre-stored knowledge to solve complex SDMs and fine-tune it via Proximal Policy Optimization (PPO) (Schulman et al., 2017). Our variant, called GFlan-Prior, incorporates a KL regularizer into the PPO loss to optimize Eq. 2. During implementation, we sample $k = 5$ examples from LLM prior distribution $p_{\text{LLM}}(a|s)$ to compute the approximated KL divergence. Formally, for the $i$-th iteration, the policy $\pi_{\theta_i}$ is learned by optimizing:

$$\arg \max_{\theta} \mathbb{E}_{(s_t, a_t) \sim \pi_{\theta_{i-1}}} \text{CLIP} \left( \frac{\pi_\theta(a_t|s_t)}{\pi_{\theta_{i-1}}(a_t|s_t)} \right) \hat{A}_t + \eta \mathcal{H}(\pi_\theta(\cdot|s_t)) - \alpha \text{KL}[\pi_\theta(\cdot|s_t) \| \hat{p}_{\text{LLM}}(\cdot|s_t)], \quad (8)$$

where $A_t$ is the estimated advantage, $\eta$ is the hyperparameter that encourages exploration, and $\hat{p}_{\text{LLM}}(\cdot|s_t)$ is the discrete distribution constructed by $k$ action proposals from the LLM. $\theta_{i-1}$ represents the parameters of the action policy learned in the last iteration. The illustration of the training process of GFlan-Prior can be found in Figure 2.

## 4 EXPERIMENTS

In this section, we demonstrate the effectiveness of incorporating the LLM prior into the RL framework, including both policy-based and value-based algorithms in online and offline settings. LLM priors are used to reduce the exploration and optimization space or regulate the action policy's behavior, thereby resulting in a more efficient RL framework. Further details on experimental settings and more ablation study results can be found in the Appendix 9.

### 4.1 ENVIRONMENTS

We consider three environments:
**ALFWorld** (Shridhar et al., 2020) is a popular benchmark for examining LLM-based agents' decision-making ability. This benchmark contains thousands of Textworld (Côté et al., 2019) games with the embodied engine. We focus solely on the text game part, where the action space consists of high-level plans such as "go to a room". The legal actions are finite but large, with the maximum possible admissible action space reaching up to 50, making it challenging to explore from scratch. It contains thousands of tasks, making testing the generalization performance on unseen tasks convenient. We consider two classes of ALFWorld tasks: ALFWorld(pick) and ALFWorld(examine). There are no auxiliary rewards, except for a reward of 1.0 for reaching the final goal.
**Overcooked** We use the partially observed text overcooked game(Tan et al., 2024); the observation describes the position of visible items, and the agent should take a sequence of actions to deliver a dish. We consider two overcooked tasks: Overcooked(Tomato), deliver a dish of chopped tomato; Overcooked(Salad), deliver a salad containing chopped tomato and lettuce. The maximum possible action space is 8. Besides the reward of 1 for successfully delivering a dish, the textual Overcooked environment from (Tan et al., 2024) also provides dense reward signals. The reward shaping is as follows: 0.2 for correctly chopping an ingredient, 1 terminal reward for successfully delivering the correct dish, $-0.1$ for delivering any incorrect item, and $-0.001$ for each time step.
**Frozen Lake** is a grid world game; the agent should move to the goal position while avoiding falling into holes. There are four admissible actions: up, down, right, and left. The agent will receive a reward of 1 for reaching the final goal and a penalty of $-1$ for falling into holes. More details of state and action representations can be found in Appendix 9.1.

### 4.2 BASELINES

We first test the ability of **LLM Prior** to solve SDMs in a zero-shot manner, without deliberately designed prompts. Next, we will introduce the value-based and policy-based RL algorithms:

**Valued-Based RL:** For the online setting, we use **DQN** and our **DQN-Prior**, which explores and updates the Q-function within the LLM prior space. For the offline setting, we compare the **CQL**, **CQL-prior**, and **Behavioral Cloning(BC)**. Similarly, compared to CQL, CQL-Prior regulates and updates Q-values in the narrowed prior action space. We set $k = 5$ for DQN-Prior and CQL-Prior. The BC utilizes the Flan-T5 small (Rae et al., 2021) as the action policy, and learns the action policy by optimizing: $\arg \min_{\pi} -\mathbb{E}_{(s,a) \sim \mathcal{D}}[\log \pi(a|s)]$.

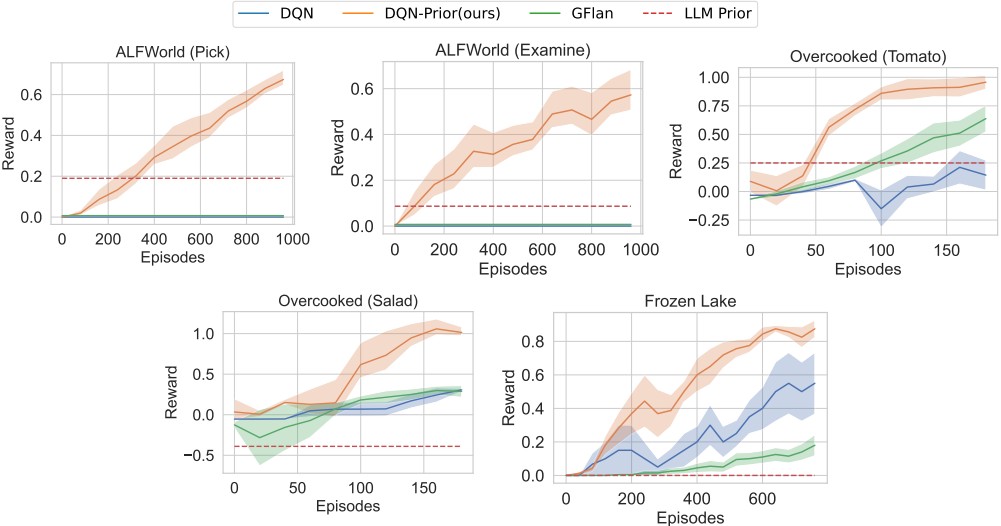

Figure 3: Results of comparison with online baselines. We plot the mean and standard error of the cumulative reward. Periodic evaluation results are plotted for online baselines.

**Policy-Based RL: GFlan** (Carta et al., 2023) lets the LLM Flan-T5 small as the action policy and finetunes it via PPO (Schulman et al., 2017). **GFlan-Prior** adds a KL constraint between the optimized action policy and LLM prior action on the basis of GFlan, as shown in Eq. 26.

**LLM Setting** Due to the Qwen-1.5 series (Bai et al., 2023) containing a range of LLMs with different scales, we use Qwen-1.5 for our main experiments. Specifically, we use Qwen-1.5 7B (Bai et al., 2023) as the backbone of the LLM prior for most environments, except for the more complex Overcooked(Salad) task, where Qwen-1.5 14B is used.

### 4.3 RESULTS ANALYSIS

#### 4.3.1 INCORPORATING LLM PRIORS INTO VALUE-BASED RL

**The effectiveness of LLM Prior on online Q-learning** As shown in Figure 3, our DQN-Prior outperforms traditional online RL baselines such as DQN and GFlan by exploring and optimizing within a narrowed, more valuable prior action space. In complex ALFWorld tasks, DQN and GFlan struggle to bootstrap due to the large original action space, which can include up to 50 possible actions and the lack of dense reward incentives. In contrast, our DQN-Prior leverages the LLM's domain knowledge to reduce the exploration space to just five suboptimal actions, significantly improving sample efficiency. Additionally, in the Overcooked (Salad) and Frozen Lake tasks, the LLM prior cannot directly solve these tasks by sampling one action per state. However, DQN-Prior still achieves success and demonstrates greater sample efficiency than traditional RL methods, despite the small action space (only 4 or 8 actions). This shows that the LLM prior can provide a suboptimal action proposal space, and conducting RL within this space significantly enhances sample efficiency.

Table 1: Results of offline algorithms. We consider two tasks, ALFWorld (Pick) and Overcooked (Salad), referred to as Pick and Salad for simplicity. To further compare the sample efficiency of the baselines, we use offline datasets of varying sizes for Overcooked (Salad), denoted as Salad (N), each containing approximately N $(s, a, s')$ tuples.

| Baseline | Pick | Salad(1000) | Salad(4000) | Salad(8000) | Salad(12000) | Salad(24000) |
|---|---|---|---|---|---|---|
| DQN | 0.27 | 0.14 | 0.31 | 0.32 | 0.32 | 0.32 |
| CQL | 0.03 | 0.32 | 0.32 | 0.32 | 0.32 | **1.33** |
| DQN-Prior | 0.49 | 0.98 | 0.81 | - | - | - |
| CQL-Prior | **0.80** | **1.01** | **1.19** | - | - | 1.31 |
| BC | 0.31 | 0.57 | 0.91 | 0.78 | 0.53 | 0.74 |

Table 2: Results on the generalization performance of posterior sampling on ALFWorld(Pick). We use Qwen-1.5 7B with $k = 5$ to generate LLM prior action proposals for training the Q function via DQN-Prior and CQL-Prior. This LLM prior configuration is highlighted by (*).

| | Online (DQN-Prior) | | | | | | | |
|---|---|---|---|---|---|---|---|---|
| | **Seen Tasks** | | | | **Unseen Tasks** | | | |
| | $k=1$ | $k=5$ | $k=10$ | $k=15$ | $k=1$ | $k=5$ | $k=10$ | $k=15$ |
| Qwen-1.5 4B | 0 | 0.35 | **0.46** | 0.35 | 0 | 0.08 | 0.25 | **0.29** |
| Qwen-1.5 7B | 0.19 | 0.62* | **0.77** | 0.65 | 0.04 | 0.38 | **0.42** | **0.42** |
| LLaMa-3 8B | 0.04 | 0.62 | **0.73** | 0.69 | 0.13 | **0.42** | 0.38 | 0.38 |
| Qwen-1.5 14B | 0.19 | 0.65 | **0.77** | 0.65 | 0.16 | **0.54** | **0.54** | 0.38 |
| Qwen-1.5 32B | 0.35 | 0.77 | **0.81** | 0.73 | 0.33 | **0.5** | **0.5** | 0.46 |
| | Offline (CQL-Prior) | | | | | | | |
| | **Seen Tasks** | | | | **Unseen Tasks** | | | |
| | $k=1$ | $k=5$ | $k=10$ | $k=15$ | $k=1$ | $k=5$ | $k=10$ | $k=15$ |
| Qwen-1.5 4B | 0 | 0.27 | 0.38 | **0.62** | 0.0 | 0.17 | 0.17 | **0.38** |
| Qwen-1.5 7B | 0.19 | 0.80* | **0.81** | 0.77 | 0.04 | 0.46 | **0.50** | **0.50** |
| LLaMa-3 8B | 0.04 | 0.85 | **0.88** | 0.81 | 0.13 | **0.54** | 0.50 | **0.54** |
| Qwen-1.5 14B | 0.19 | 0.73 | **0.92** | 0.73 | 0.16 | 0.54 | **0.58** | 0.54 |
| Qwen-1.5 32B | 0.35 | 0.81 | **0.85** | 0.69 | 0.33 | **0.63** | 0.42 | 0.50 |

**The effectiveness of LLM Prior on offline Q-learning**  Table 1 illustrates the performance of baselines trained on offline datasets. A more detailed description of the offline datasets can be found in the Appendix. Table 1 shows that CQL-Prior outperforms all other baselines on the ALFWorld (Pick) and Overcooked (Salad) datasets. Although DQN-Prior lacks constraints on Q-values, it still demonstrates superior performance compared to DQN and CQL, which operate on the full action space. These results indicate that in offline RL, avoiding overestimation of Q-values and performing Bellman updates only within the suboptimal prior action space reduces optimization complexity, thereby improving sample efficiency.

Additionally, we provide further evidence to support this conclusion by testing the sample requirements for CQL and CQL-Prior. We find that traditional CQL, with fewer than 12000 examples, fails to learn how to achieve the final goal, only managing to reach the subgoal of chopping the tomato and lettuce (with a reward of $0.4$). In contrast, CQL-Prior, with just around 1000 examples, successfully learns how to achieve the final goal (with a reward of $1.0$) with high probability. Therefore, our CQL-Prior reduces the number of required samples by at least 90% compared to CQL on this task.

**The generalization ability across unseen tasks and LLMs**  Table 2 demonstrates the generalization ability of our posterior sampling framework by combining the Q-function, trained with the LLM prior setting of Qwen-1.5 7B and $k = 5$, with other LLMs and varying action proposal numbers ($k$) during inference. Even though the Q-network is trained with Qwen-1.5 7B and $k = 5$, it can still effectively guide the decision-making process, i.e., action selection, for unseen LLMs of different scales and architectures. It is worth noting that, although Qwen-1.5 4B was originally unable to solve ALFWorld (Pick), it becomes capable of tackling such complex tasks by leveraging the Q-function trained with more powerful LLMs. This value-based RL approach, which utilizes an LLM prior, presents a promising opportunity to train an information-rich Q-network using larger LLMs while deploying only the Q-network and smaller LLMs on the client side. This method enhances reasoning speed and reduces deployment complexity. Furthermore, larger LLMs generally perform better during inference, as their action proposals are of higher quality. $k = 1$ indicates the performance of the LLM on its own, without involving the Q-function. The setting of $k = 5$ was chosen during training for querying efficiency, but it is not necessarily the optimal choice. We observed better test performance with $k = 10$ on seen tasks, indicating that our method can generalize to a larger action space to some extent. In most cases, inference performance increases and then decreases as $k$ increases. This may occur because a larger action space is more likely to include the optimal action, but an excessively large action space can introduce unseen $(s, a)$ pairs during training, leading to out-of-distribution (OOD) problems in both online and offline settings.

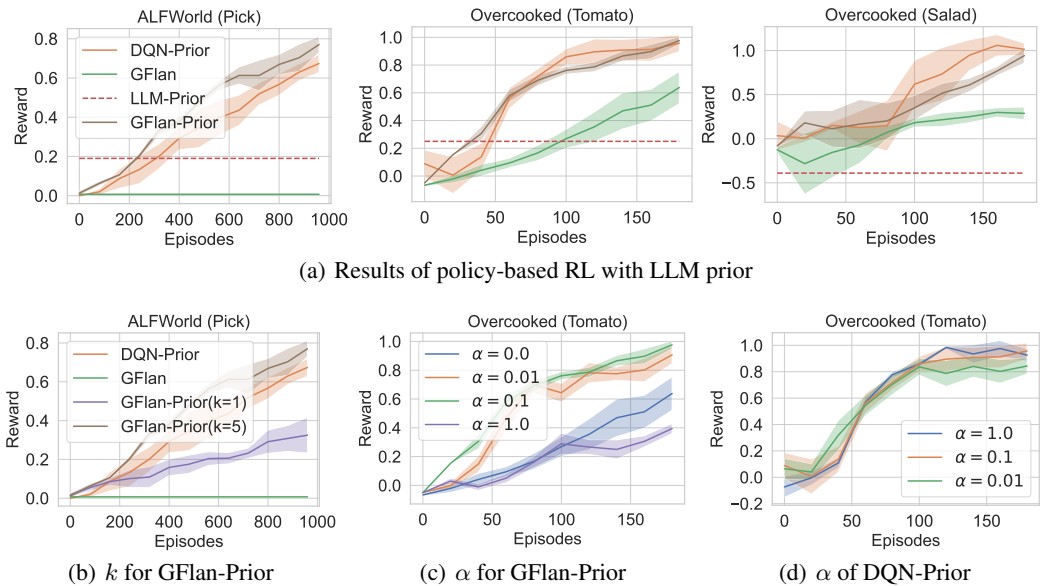

Figure 4: (a) Comparison of policy-based RL algorithms, we plot DQN-Prior and LLM-Prior for reference. (b) The ablation study on the number of LLM action proposals $k$ used to approximate the KL divergence. (c) The ablation of the KL coefficient of GFlan-Prior. (d) The ablation of the softmax temperature over Q values of the DQN-Prior

Regarding generalization to unseen tasks, multiple LLM action proposals with $k > 1$, guided by the Q-function, generally outperform the LLM's zero-shot performance with $k = 1$. This demonstrates the generalization capability of our posterior sampling framework, enabled by the powerful LLM as an action prior and the Q-function's ability to compress interactive experiences.

### 4.3.2 INCORPORATING LLM PRIORS INTO POLICY-BASED RL

The Figure 4(a), 4(b), 4(c) show the results of policy-based RL algorithms. As shown in Figure 4(a), GFlan-Prior significantly outperforms GFlan and achieves performance comparable to DQN-Prior. Unlike GFlan, GFlan-Prior leverages the prior knowledge embedded in LLMs by aligning with the LLM prior distribution through the incorporation of a KL constraint, as shown in Equation 26. In our main results, we sample $k = 5$ action proposals from the LLM prior to approximate the KL divergence. An ablation study on $k$ can be found in Figure 4(b) and Figure 5 in the Appendix. Similar to the results in Table 2, with $k = 5$, we are more likely to sample the optimal action than with $k = 1$, thereby better supervising the learning of the action policy.

The ablation study on the coefficient of the KL divergence is shown in Figure 4(c). A large $\alpha$ will cause the learned policy $\pi$ to closely follow the LLM prior proposals, while a small $\alpha$ encourages exploration. In DQN-Prior, the hyperparameter $\alpha$ regulates exploration by acting as the temperature in the softmax over Q, generating a Boltzmann distribution based on Q, as investigated in Figure 4(d). GFlan-Prior appears to be more sensitive to the hyperparameter on controlling exploration than DQN-Prior.

## 5 RELATED WORK

This study explores the fusion of Bayesian inference and LLMs for decision-making tasks. Herein, we present related works on LLM-Agent for decision-making tasks and how previous research pursues this combination. More discussions on the use of LLMs for value-based inference and reward function generation are provided in Appendix 10.

**LLM-driven Agent**   The idea of LLM-as-Agent has gone viral since the release of powerful large language models. Among all the works, some apply direct prompt engineering techniques

using human-designed prompt and retrieval to harness LLMs for complex interactive tasks, such as ProAgent by (Zhang et al., 2023), Voyager by (Wang et al., 2023) and generative agent by (Park et al., 2023). Many researchers have also tried searching-based methods. (Yao et al., 2023) propose Tree-of-Thought (TOT) using BFS/DFS algorithm to search accurate decision sequences. (Hao et al.) suggests using Monte-Carlo Tree Search to perform a more comprehensive search. These methods often fall into the framework of Proposer-Verifier (Snell et al., 2024) where the language proposes a number of possible sequences and the verifier picks desirable candidates. There are also a few works that relate LLM to conventional reinforcement learning settings and investigate how the superiority in language space helps RL agents' capability in decision-making. For instance, (Brooks et al., 2024) uses LLM as a world model and performs policy iteration by in-context learning. GFlan proposed by (Carta et al., 2023) uses a learnable LLM as a probability likelihood estimator for all possible actions and incorporates it into actor-critic learning frameworks where a value function generates assessment and fine-tuning the LLM correspondingly. (Yan et al., 2023) also resort to an actor-critic learning paradigm but instead of fine-tuning the language model, they train a simple classifier that selects valuable outputs from a fixed language model (e.g. GPT3.5). (Zhang et al., 2024b) use LLM as a behavior regularize and add it to the value estimate. (Wen et al., 2024) leverages entropy-regularized reinforcement learning to train LLM agents for verbal sequential decision-making tasks. Our work also falls into the category of LLM-driven RL agent. However, different from others, we see a large language model as an excellent action proposer and perform Q-learning directly on the reformed action sequence space, thereby achieving better learning efficiency with minimum modification.

**LLM in Bayesian Sense**    The probabilistic nature of transformer-based language models makes them well-suited for a Bayesian inference framework. As highlighted by (Korbak et al., 2022), the Bayesian approach offers a unified perspective on both modeling and inference. In in-context learning (ICL), some researchers argue for the Bayesian properties of ICL ((Jiang, 2023; Wang et al., 2024b; Ye et al., 2024)). Extending beyond ICL, (Yang & Klein, 2021) trains a classifier to approximate the likelihood function, framing conditional text generation as a Bayesian inference problem, with the LLM serving as the prior. (Hu et al.) interprets Chain-of-Thought (CoT) reasoning in natural language processing (NLP) tasks, such as text infilling and sequence continuation, as probabilistic inference problems, using GFlowNet to fine-tune the model. In this context, the language model functions as a unified probabilistic generative process, capable of representing both prior and likelihood distributions. Similarly, (Zhao et al., 2024b) adopts a Bayesian perspective, utilizing Sequential Monte Carlo (SMC) methods to generate undesired outcomes. (Gallego, 2024) frames LLM inference as sampling from a posterior distribution focused on high-reward regions, applying this approach to self-improvement tasks. Inspired by these works, we also formulate our agent framework within a Bayesian setting, exploring the use of LLMs as priors.

## 6   CONCLUSION

In this work, we aim to harness the rich domain knowledge and strong reasoning abilities of LLMs to tackle complex decision-making tasks while avoiding the costly processes of prompt design and fine-tuning. Given their extensive prior knowledge but limited task-specific experience, we do not rely on LLMs to directly generate optimal plans. Instead, we simplify their role to producing reliable action proposals. To this end, we treat the powerful LLM as an action prior distribution and, from a Bayesian inference perspective, examine how to integrate LLM priors into solving MDPs. We employ variational inference and direct posterior sampling to achieve this, ultimately proposing both policy-based and value-based RL frameworks incorporating LLM priors. These priors enhance traditional RL frameworks by narrowing the exploration space or guiding the action policy through suboptimal LLM-generated proposals. Experimental results demonstrate a substantial improvement in sample efficiency when incorporating fixed LLM priors into RL frameworks in both online and offline settings. This work focuses specifically on text-based games with finite action spaces. In future research, we plan to explore scenarios with free-form and infinite action spaces and investigate deliberate reasoning approaches to further refine the quality of LLM-generated action proposals.

## ACKNOWLEDGEMENTS

Co-author Haifeng Zhang thanks IPT Project 2024002.

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

# 7 DESCRIPTION OF CLASSIC Q-LEARNING ALGORITHM

We introduce the classic Q-learning algorithm in both offline and online settings.

**Q-learning**   Q-learning is a classic value-based RL algorithm, which iteratively applies the Bellman optimal operator to train the Q-function, formally written as $\mathcal{B}^* Q(s,a) = r(s,a) + \gamma \mathbb{E}_{s' \sim P(\cdot|s,a)} \left[ \max'_a Q(s',a') \right]$. After the optimal Q-function $Q^*$ learned, then the optimal action policy respect to the $Q^*$ is given as: $\pi^*(\cdot|s) = \arg\max_a Q^*(s,a)$. DQN (Mnih, 2013) is a well-known Q-learning algorithm, which can process various classes of the state information such as image and language, which learns a deep neural network $Q^\theta(s,a) \approx Q^*(s,a)$ to approximate $Q^*(s,a)$ and uses TD-learning to iteratively update the Q network, the loss function for the $i$-the iteration is given as:

$$\mathcal{L}_i(\theta) = \mathbb{E}_{(s,a,s') \sim \mathcal{D}} \left[ (Q^{\theta_i}(s,a) - y_i)^2 \right], \tag{9}$$

where $\mathcal{D}$ is the reply buffer, $y_i = \mathcal{B}^* Q^{\theta_{i-1}}(s,a) = r(s,a) + \gamma \max_{a' \in \mathcal{A}} Q^{\theta_{i-1}}(s',a')$. $\theta_{i-1}$ is the learned parameters of the last iteration and is frozen during the gradient descent for optimizing the loss function $\mathcal{L}_i$.

**Conservative Q-learning**   CQL (Kumar et al., 2020) framework is designed for offline reinforcement learning, where only offline data is available, but no further online exploration is accessible. Different from traditional Q-learning, CQL introduces an additional regularization term on top of the standard DQN framework or actor-critic framework(such as SAC (Haarnoja et al., 2018)), ensuring that the learned Q-function lower-bounds the actual value. The lower-bounded Q-values help mitigate the overestimation problem commonly encountered with out-of-distribution (OOD) state-action pairs. The CQL loss on the top of value-based DQN is given as:

$$\mathcal{L}_i(\theta) = \beta \mathbb{E}_{(s,a) \sim \mathcal{D}} \left[ \log \sum_{a' \in \mathcal{A}} \exp(Q^{\theta_i}(s,a')) - Q^{\theta_i}(s,a) \right] + \frac{1}{2} \mathbb{E}_{(s,a,s') \sim \mathcal{D}} \left[ (Q^{\theta_i}(s,a) - y_i)^2 \right], \tag{10}$$

where $y_i = r(s,a) + \gamma \max_{a'} Q^{\theta_{i-1}}(s',a')$ and $\beta$ is a hyper-parameter.

# 8 PROOFS

## 8.1 PROBABILISTIC INFERENCE DERIVATION

In this section, we derive the detailed formulation for probabilistic inference for using LLM as a prior. This includes the policy-based method where we learn a parametrized policy function and the value-based method where we directly perform posterior inference.

**Variational Inference**   Assume that we have the optimality variable $\mathcal{O}$ indicating the quality of a trajectory $\tau = \{s_0, a_0, r_0, s_1, ..., s_n\}, p(\tau) = p(s_0) \prod_t p(a_t|s_t) p(s_{t+1}|s_t, a_t)$. The likelihood function with temperature parameter $\alpha$ can be written as:

$$p(\mathcal{O} = 1|\tau) = \exp\left( \sum_t \gamma^t r_t / \alpha \right) \tag{11}$$

Our goal is to maximize the optimal marginal distribution formulated as follows:

$$\begin{aligned} p(\mathcal{O} = 1) &= \int p(\mathcal{O} = 1, \tau) d\tau \\ &= \int q(\tau) \frac{p(\mathcal{O} = 1|\tau) p(\tau)}{q(\tau)} d\tau \end{aligned} \tag{12}$$

where $q(\tau)$ is a parametrized variational distribution. Then, our variational inference objective can be written as:

$$\begin{aligned} \log p(\mathcal{O} = 1) &\geq \int q(\tau) \log \left[ p(\mathcal{O} = 1|\tau) \frac{p(\tau)}{q(\tau)} \right] d\tau \\ &= \mathbb{E}_{q(\tau)} \left[ \log p(\mathcal{O} = 1|\tau) \right] - \mathbf{KL} \left[ q(\tau) | p(\tau) \right] \\ &= \mathbf{ELBO} \end{aligned} \tag{13}$$

In the context of language models, we define prior $p(\tau)$ as trajectory distribution generated through a fixed LLM denoted as:

$$p_{LLM}(\tau) = p(s_0) \prod_t p_{LLM}(a_t|s_t)p(s_{t+1}|s_t, a_t) \tag{14}$$

where $p_{LLM}(a_t|s_t)$ is the probability likelihood of the generated output $a_t$ given input text $s_t$. The variational distribution $q(\tau)$ can also be decomposed in the same way as:

$$q(\tau) = p(s_0) \prod_t \pi_\theta(a_t|s_t)p(s_{t+1}|s_t, a_t) \tag{15}$$

where the parametrized policy function $\pi_\theta$ can be a learnable language model with parameter $\theta$. Substituting these factorizations back to ELBO we have the step-wise objective function written as:

$$-\mathcal{L} = \sum_t \mathbb{E}_{\pi_\theta} \left[\gamma^t r_t\right] - \alpha \mathbf{KL} \left[\pi_\theta(a_t|s_t) \| p_{LLM}(a_t|s_t)\right] \tag{16}$$

$$\pi^* = \arg\min_\theta \mathcal{L} \tag{17}$$

**Direct Posterior Inference**   Instead of learning a parametrized policy which might involve fine-tuning a language model, we can also try directly sampling from the posterior distribution formulated as:

$$\begin{aligned} p(\tau|\mathcal{O} = 1) &= \frac{p(\mathcal{O} = 1|\tau)p(\tau)}{p(\mathcal{O} = 1)} \\ &\propto p(\mathcal{O} = 1|\tau)p(\tau) \\ &= \prod_t p(\mathcal{O}_t = 1|s_t, a_t)p_{LLM}(a_t|s_t)p(s_{t+1}|s_t, a_t) \end{aligned} \tag{18}$$

in which we define $p(\tau)$ as in Equation 14. According to Control-as-Inference framework proposed by Levine (2018), we can formulate posterior inference as Soft Q-Learning where Q-values are updated using a modified soft Bellman equation:

$$\begin{aligned} Q^\pi(s_t, a_t) &= r(s_t, a_t) + \gamma \mathbb{E}_{s_{t+1} \sim p(\cdot|s_t, a_t)} \left[V(s_{t+1})\right] \\ V(s_t) &= \log \int \exp(Q(s_t, a_t)/\alpha)da_t \end{aligned} \tag{19}$$

and the policy can be defined as a softmax function over the Q-values: $\pi(a|s) = \exp(Q^\pi(s, a)/\alpha)/\int_{a'} \exp(Q^\pi(s, a')/\alpha)da'$, ensuring that actions with higher Q-values are more likely to be chosen. In our paper, however, due to the enormously large space of (textual) state-action pair, we also apply sampling approximation methods. Specifically, we leverage similar ideas from Fourati et al. (2024) which sample a random subset of actions from the complete action space and seek the optimal within this subset. In our case, we rely on a fixed LLM to provide such action subspace and can further narrow down the space through repeated sampling (Brown et al. (2024)):

$$\begin{aligned} \pi(a|s) &= \frac{\mathbf{1}_{a \in p_{LLM}(\cdot|s)} \cdot \exp\left(Q^\pi(s, a)/\alpha\right)}{\int_{a'} \mathbf{1}_{a \in p_{LLM}(\cdot|s)} \cdot \exp\left(Q^\pi(s, a')/\alpha\right) da'} \\ &\approx \mathbf{1}_{a \in \{a^i\}} \cdot \frac{1}{N} \frac{\exp(Q^\pi(s, a^i)/\alpha)}{\sum_i \exp(Q^\pi(s, a^i)/\alpha)}, a^i \sim p_{LLM}(a|s) \end{aligned} \tag{20}$$

This means we restrict the Bellman backup to a specific action subspace that can potentially provide a refined set of actions based on context, largely reducing computational complexity and focusing on more relevant actions.

## 8.2   PROOF OF PROPOSITION 1

**Proposition 1.** *The above action inference strategy—selecting an action from the action priors reweighted based on Q-values—can be described as following a distribution q. As $k \to \infty$, we have:*

$$\lim_{k \to \infty} q(a|s_t) = p_{LLM}(a|s_t)\exp(Q^\theta(s, a)/\alpha)/\mathbb{E}_{a_j \sim p_{LLM}(\cdot|s)} \exp(Q^\theta(s, a_j)/\alpha) \tag{4}$$

*The limiting policy corresponds to the policy that optimizes the Q-values with a KL regularizer:*

$$\lim_{k \to \infty} q(\cdot|s_t) = \arg\max_{\pi} \mathbb{E}_{\pi(a|s_t)}[Q^{\theta}(s_t, a)] - \alpha KL\left(\pi(\cdot|s_t) \| p_{LLM}(\cdot|s_t)\right) \tag{5}$$

*Then, the posterior sampling strategy is highly related to the solution of variational inference as shown in Eq. 2. Proof. Please see the appendix 8.2.*

*Proof.* The proof mainly follows the (Li et al., 2024). Denote the above sampling strategy indeed follows a distribution $q$, and for each action $a$, the probability of $a$ is sampled from the distribution $q$ is given by:

$$
\begin{aligned}
q(a|s_t) &= \mathbb{E}_{\{a_1,\cdots,a_k\} \sim p_{\text{LLM}}(\cdot|s)} \left[ \sum_{i=1}^{k} \mathbb{I}(a_i = a) \frac{\exp(Q^{\theta}(s_t, a_i)/\alpha)}{\sum_{j=1}^{k} \exp(Q^{\theta}(s_t, a_j)/\alpha)} \right] \\
&= \mathbb{E}_{\{a_1,\cdots,a_k\} \sim p_{\text{LLM}}(\cdot|s)} \left[ \frac{\sum_{i=1}^{k} \mathbb{I}(a_i = a)}{\sum_{j=1}^{k} \exp(Q^{\theta}(s_t, a_j)/\alpha)} \right] \exp(Q^{\theta}(s_t, a)/\alpha) \\
&= \mathbb{E}_{\{a_1,\cdots,a_k\} \sim p_{\text{LLM}}(\cdot|s_t)} \left[ \frac{\frac{1}{k}\sum_{i=1}^{k} \mathbb{I}(a_i = a)}{\frac{1}{k}\sum_{j=1}^{k} \exp(Q^{\theta}(s_t, a_j)/\alpha)} \right] \exp(Q^{\theta}(s_t, a)/\alpha)
\end{aligned}
\tag{21}
$$

According to the Law of Large Numbers, when $k \to \infty$, we have:

$$
\begin{aligned}
\lim_{k \to \infty} q(a|s_t) &= \lim_{k \to \infty} \mathbb{E}_{\{a_1,\cdots,a_k\} \sim p_{\text{LLM}}(\cdot|s_t)} \left[ \frac{\frac{1}{k}\sum_{i=1}^{k} \mathbb{I}(a_i = a)}{\frac{1}{k}\sum_{j=1}^{k} \exp(Q^{\theta}(s_t, a_j)/\alpha)} \right] \exp(Q^{\theta}(s_t, a)/\alpha) \\
&= \lim_{k \to \infty} \mathbb{E}_{\{a_1,\cdots,a_k\} \sim p_{\text{LLM}}(\cdot|s_t)} \left[ \frac{p_{\text{LLM}}(a|s_t)}{\mathbb{E}_{a_j \sim p_{\text{LLM}}(\cdot|s_t)} \exp(Q^{\theta}(s_t, a_j)/\alpha)} \right] \exp(Q^{\theta}(s_t, a)/\alpha) \\
&= p_{\text{LLM}}(a|s_t) \frac{\exp(Q^{\theta}(s, a)/\alpha)}{\mathbb{E}_{a_j \sim p_{\text{LLM}}(\cdot|s)} \exp(Q^{\theta}(s, a_j)/\alpha)}
\end{aligned}
\tag{22}
$$

Following the proof process from the Appendix A.1. in Rafailov et al. (2023), we have:

$$\arg\max_{\pi} \mathbb{E}_{\pi(a|s_t)}[Q^{\theta}(s_t, a)] - \alpha\text{KL}\left(\pi(\cdot|s_t) \| p_{\text{LLM}}(\cdot|s_t)\right) = p_{\text{LLM}}(a|s_t) \frac{\exp(Q^{\theta}(s, a)/\alpha)}{\mathbb{E}_{a_j \sim p_{\text{LLM}}(\cdot|s)} \exp(Q^{\theta}(s, a_j)/\alpha)} \tag{23}$$

$\square$

Additionally, as shown in Sec. 2, the variational inference approach learns the optimal policy $\pi$ by maximising:

$$\arg\max_{\pi} \sum_{t} \mathbb{E}_{\pi}\left[\gamma^t r_t\right] - \alpha\,\text{KL}(\pi(a|s_t) \| p_{\text{LLM}}(a|s_t)). \tag{24}$$

Introducing the occupancy measure $\rho$, $\rho(s) = \frac{1}{1-\gamma}\sum_{t=0}^{\infty}[\gamma^t \mathbb{P}(s_t = s|a \sim \pi)]$, we have the following form objective respected to the Q-function

$$\arg\max_{\pi} \mathbb{E}_{\rho(s)}[\mathbb{E}_{a \sim \pi}[Q^{\pi}(s, a)] - \alpha\text{KL}(\pi(a|s_t) \| p_{\text{LLM}}(a|s_t))] \tag{25}$$

Combining Equations 23 and 25, we find that the variational inference approach and the direct posterior sampling yield similar solutions.

# 9 ADDITIONAL EXPERIMENT DETAILS

## 9.1 ENVIRONMENTS

We consider three environments: ALFWorld, Overcooked and Frozen Lake.
**ALFWorld** We consider two classes of ALFWorld tasks: ALFWorld(pick) and ALFWorld(examine). For ALFWorld (Pick), we evaluate the online training baselines on 28 tasks, such as "put the cellphone on the armchair," and test the generalization ability on 26 unseen tasks. For ALFWorld (Examine),

|  | ALFWorld(Pick) | ALFWolrd(Examine) | Cook(Tomato) | Cook(Salad) | Frozen Lake |
|---|---|---|---|---|---|
| Horizon | 60 | 60 | 30 | 50 | 20 |

Table 3: Maximize horizon of environments.

we use 11 tasks, such as "examine the laptop with the desk lamp." There are no auxiliary rewards, except for a reward of $1.0$ for reaching the final goal.

Examples of observations and admissible action descriptions for three environments are shown below:

---

**For ALFWorld(Pick)**

**Observation:** Current observation:You are in the middle of a room. Looking quickly around you, you see a armchair 1, a bed 1, a diningtable 1, a drawer 2, a drawer 1, a garbagecan 1, a sidetable 2, and a sidetable 1.
Your task is to: put some cellphone on armchair..
**Admissible actions** You are allowed to take the following actions: go to armchair 1, go to bed 1, go to diningtable 1, go to drawer 1, go to drawer 2, go to garbagecan 1, go to sidetable 1, go to sidetable 2, inventory, look.

---

**For Overcooked(Tomato)**

**Observation:** Your task is to serve the dish of a bowl only containing chopped tomato. There is a fixed cutting board in the room. You notice a tomato on the table. Currently you don't have anything in hand.
**Admissible actions** You are allowed to take the following actions: pick up the tomato, take the bowl, walk to the cutting board, serve nothing, chop nothing.

---

**For Frozen Lake**

**Observation:** You are in a 4x4 grid where each cell is either a frozen cell (F) indicating a safe position, You must avoid to reach a hole (H), and the goal (G) that we aim to reach.
Input: Currently the agent is at position (0, 0), which direction need to take?
Admissible actions: Go Left leads to position (0, 0), start position. Go Right leads to position (0, 1), a frozen cell (F). Go Up leads to position (0, 0), start position. Go Down leads to position (1, 0), a frozen cell (F).
**Admissible actions** You are allowed to take the following actions: Go Left, Go Right, Go Up, Go Down.

---

### 9.2 LLM PRIOR IMPLEMENTATION

There are two ways to sample an action $a$ from the LLM prior $p_{\text{LLM}}(\cdot|s_t)$. First, since the state and action can be described as text, and assuming the action $a$ consists of $k$ tokens, the probability of the LLM generating action $a$ is given by $p(a|s_t) = \prod_{i=1}^{k} p_{\text{LLM}}(a_i|s_t, a_{<i})$. Based on this probability, the first type of LLM prior computes a distribution over actions, denoted as $p_{\text{LLM}}^{\text{dist}}(a|s_t) = \frac{\exp p(a|s_t)}{\sum_{a'} \exp p(a'|s_t)}$. In contrast, the second approach involves sampling a free-form output from the LLM, which is then mapped to an executable action via a simple rule-based projection $\mathcal{P}$. We denote this LLM prior, which relies on mapping the LLM's output, as $p_{\text{LLM}}^{\text{map}}$, where $a \sim p_{\text{LLM}}^{\text{map}}(\cdot|s_t) \leftrightarrow o \sim \text{LLM}(\cdot|s_t), a = \mathcal{P}(o)$.

We observed a performance gap between these two types of LLM action priors. For instance, in the ALFWorld pick & place task, $p_{\text{LLM}}^{\text{map}}$ achieved a winning rate of $19\%$, while $p_{\text{LLM}}^{\text{dist}}$ failed, with a winning rate of $0\%$, both using the LLM Qwen1.5-7B. Therefore, in this work, we implement $p_{\text{LLM}}^{\text{map}}$ as the LLM prior for thoughtfully exploiting the power of the LLM in SDM tasks, which we denote as $p_{\text{LLM}}$ for simplicity. An Example of LLM's output is given as:

> **For ALFWorld(Pick) from Qwen-1.5 7B**
>
> **Input:** Current observation:You are in the middle of a room. Looking quickly around you, you see a armchair 1, a bed 1, a diningtable 1, a drawer 2, a drawer 1, a garbagecan 1, a sidetable 2, and a sidetable 1.
> Your task is to: put some cellphone on armchair.. You are allowed to take the following actions: go to armchair 1, go to bed 1, go to diningtable 1, go to drawer 1, go to drawer 2, go to garbagecan 1, go to sidetable 1, go to sidetable 2, inventory, look. Please select an action from the admissible actions.
> **LLM Output to Action:** 1: take cd 2 from sidetable 2 , ——> take cd 2 from sidetable 2
> 2: Open drawer 2 Result: You open door, ——> Open drawer 2
> 3: look Description: You look around the room., ——> look
> 4: look > Current observation:You arrive at loc, ——> look
> 5: open drawer 2 Loc 8: You open drawer 2 Loc 8 ——> open drawer 2 Loc 8
> We map the LLM's output to an executable action using a simple rule-based method, extracting the executable actions directly from the LLM's output.

## 9.3 HYPER-PARAMETERS

Our algorithms are trained on one machine with 2 40G A100. based on Pytorch-GPU 2.1 and cuda12.4. Table 6,7,8,9,10 reports the main hyper-parameters of our algorithms. For all CQL-based algorithms, we set the hyperparameter $\beta$ for regulating the Q-values, as shown in Eq. 7, as $5.0$.

**Offline Datasets** The composition of offline datasets is shown in Table 11, which reports the number of $(s, a, s')$ examples.

Table 4: The hyperparameters on ALFWorld(Pick)

| Baselines | Learning Rate | Epochs | Batch Size | Update Frequency | LLM | $\alpha$ |
|---|---|---|---|---|---|---|
| DQN-Prior | 5e-4 | 4 | 128 | 5 | / | 0.01 |
| CQL-Prior | 5e-4 | 4 | 128 | 5 | / | 0.01 |
| GFlan-Prior | 1e-4 | 16 | 64 | 16 | Flan-T5 Small | 0.01 |

Table 5: The hyperparameters on ALFWorld(Examine)

| Baselines | Learning Rate | Epochs | Batch Size | Update Frequency | LLM | $\alpha$ |
|---|---|---|---|---|---|---|
| DQN-Prior | 5e-4 | 4 | 128 | 10 | / | 0.01 |
| CQL-Prior | 5e-4 | 4 | 128 | 10 | / | 0.01 |
| GFlan-Prior | 1e-4 | 16 | 64 | 16 | Flan-T5 Small | 0.01 |

Table 6: The hyperparameters on Overcooked(Tomato )

| Baselines | Learning Rate | Batch Size | Update Frequency | LLM | $\alpha$ |
|---|---|---|---|---|---|
| DQN-Prior | 5e-4 | 128 | 5 | / | 0.1 |
| CQL-Prior | 5e-4 | 128 | 5 | / | 0.1 |
| GFlan-Prior | 1e-4 | 64 | 16 | Flan-T5 Small | 0.1 |

Table 7: The hyperparameters on Overcooked(Salad)

| Baselines | Learning Rate | Batch Size | Update Frequency | LLM | $\alpha$ |
|---|---|---|---|---|---|
| DQN-Prior | 5e-4 | 128 | 5 | / | 0.1 |
| CQL-Prior | 5e-4 | 128 | 5 | / | 0.1 |
| GFlan-Prior | 1e-4 | 64 | 16 | Flan-T5 Small | 0.1 |

Table 8: The hyperparameters on Frozen Lake

| Baselines | Learning Rate | Batch Size | Update Frequency | LLM | $\alpha$ |
|---|---|---|---|---|---|
| DQN-Prior | 5e-4 | 128 | 5 | / | 0.1 |

Table 9: The hyperparameters on ALFWorld(Pick)

| Baselines | Learning Rate | Batch Size | Update Frequency | LLM | $\alpha$ |
|---|---|---|---|---|---|
| DQN-Prior | 5e-4 | 128 | 10 | / | 0.01 |
| CQL-Prior | 5e-4 | 128 | 10 | / | 0.01 |
| GFlan-Prior | 1e-4 | 64 | 16 | Flan-T5 Small | 0.01 |

Table 10: The hyperparameters on ALFWorld(Examine)

| Baselines | Learning Rate | Batch Size | Update Frequency | LLM | $\alpha$ |
|---|---|---|---|---|---|
| DQN-Prior | 5e-4 | 128 | 10 | / | 0.01 |
| CQL-Prior | 5e-4 | 128 | 10 | / | 0.01 |
| GFlan-Prior | 1e-4 | 64 | 16 | Flan-T5 Small | 0.01 |

| Dataset | Total Examples | Good Examples | Bad Examples |
|---|---|---|---|
| ALFWorld(Pick) | 6572 | 6572 | 0 |
| Salad(1000) | 1186 | 574 | 612 |
| Salad(4000) | 3892 | 2209 | 1683 |
| Salad(8000) | 7571 | 4224 | 3347 |
| Salad(12000) | 11939 | 4004 | 7935 |
| Salad(24000) | 23869 | 7191 | 16678 |

Table 11: The composition of the offline dataset

## 9.4 ADDITIONAL RESULTS

This section presents additional experiments to further demonstrate the effectiveness of our method, including: 1) Our approach, only relying on LLM querying, is more time-efficient compared to the LLM fine-tuning-based approach. 2) Our approach is adaptable to partially observed BabyAI environments. 3) The drawback of frequently querying LLMs for action priors can be mitigated through caching techniques. 4) Inspired by the success of value-based DQN-Prior, conducting policy-based RL exclusively within the prior space is also feasible and significantly improves sample efficiency.

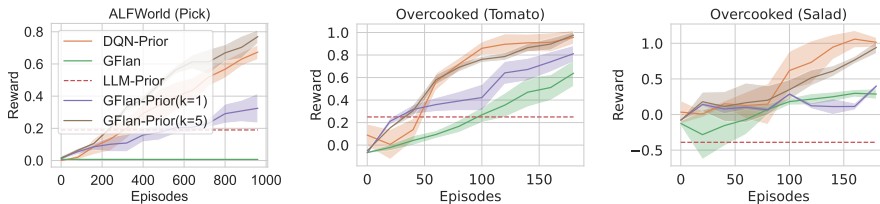

Figure 5: The ablation of the number of action proposals $k$ used for approximating KL divergence between the required action policy and the LLM prior action distribution.

**Additional ablation on $k$ for GFLan-Prior**  As shown in Figure 5, when we sample $k = 5$ action proposals from the LLM to approximate the KL divergence, it performs better than when using $k = 1$. This is because, with more proposals, the optimal action is more likely to be included in the subset, thus better guiding the action policy.

**Additional Ablation Study on the Caching Technique for Time Efficiency**  Most studies on RL fine-tuning for LLMs Tan et al. (2024); Christianos et al. (2023); Wen et al. (2024) face the

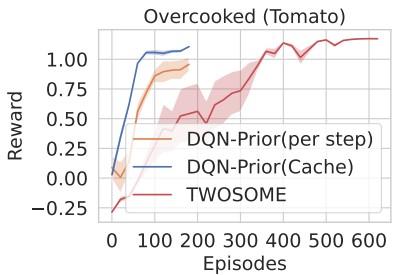

Figure 6: Comparison of DQN-Prior(Cache), DQN-Prior(query LLM per step) and TWOSOME on Overcooked (Tomato).

challenge of querying the LLM for each step. They use large LLMs (up to 7B parameters) as the action policy, and require querying the LLM at each step to collect data for online training. Since the LLM prior is fixed, we can resort to the caching technique within our framework to alleviate the need for querying the LLM at every step. We introduce a variant of the DQN-Prior algorithm, called DQN-Prior (Cache), which uses cached action candidates for previously seen observations and queries the LLM prior, storing its outputs only for new observations. We present the time and computational costs of our DQN-Prior(query LLM each time) and DQN-Prior (Cache), as well as the RLFT baseline, TWOSOME Tan et al. (2024), for learning to converge on Overcooked (Tomato) in Table 12. Results of baselines are shown in Figure 6. TWOSOME fine-tunes the LLM (Qwen-1.5 7B for a fair comparison) using the LoRA technique Hu et al. (2021) and the PPO algorithm. The time-consuming of our DQN-Prior(Cache) is significantly lower than the TWOSOME, while maintaining comparable performance. The caching technique is well-suited to our framework, as the 7B LLM is capable of providing a reliable sub-action space. DQN-Prior (Cache) significantly reduces query time costs but also achieves superior performance compared to DQN-Prior (query LLM per step), as frequent queries introduce more uncertainty.

Table 12: The training costs of DQN-Prior (Cache), DQN-Prior (query LLM per step), and TWO-SOME on Overcooked (Tomato) are reported for one training seed.

| Baselines | Training Time(min) | GPU Usage | Query LLM(Qwen-1.5 7B) Times |
|---|---|---|---|
| DQN-Prior(per step) | 31.1 | 28298 MB | 2933 |
| DQN-Prior(Cache) | 13.8 | 28340 MB | 434 |
| TWOSOME | 60.4 | 25014 MB | 5000 |

**Additional Experiment on BabyAI** We compare our policy-based algorithm, GFlan-Prior, with GFlan in the text-based BabyAI environment Carta et al. (2023), which features partial observability and random map initialization for each episode. We focus on the BabyAI task 'Go to the red ball' and set the maximum horizon at 30. The comparison results are presented in Figure 7. Our GFlan-Prior, which integrates both LLM prior knowledge and environmental interaction experience, successfully accommodates the partially observed BabyAI and outperforms both GFlan and the LLM-prior.

**Ablation Study on Using a Large LLM for the Q-Network** We conducted an experiment on Overcooked(Tomato) to compare the effects of using an LLM as the action prior versus using it to encode state-action pairs. We evaluated three baselines:

- DQN-Prior: Uses Qwen-1.5 7B to generate a sub-action space and BERT to encode (s, a) pairs for Q-value estimation.

- DQN (BERT): Uses BERT to encode (s, a) pairs and performs DQN across the full action space.

- DQN (Qwen-1.5 7B): Uses Qwen-1.5 7B to encode (s, a) pairs and performs DQN across the full action space.

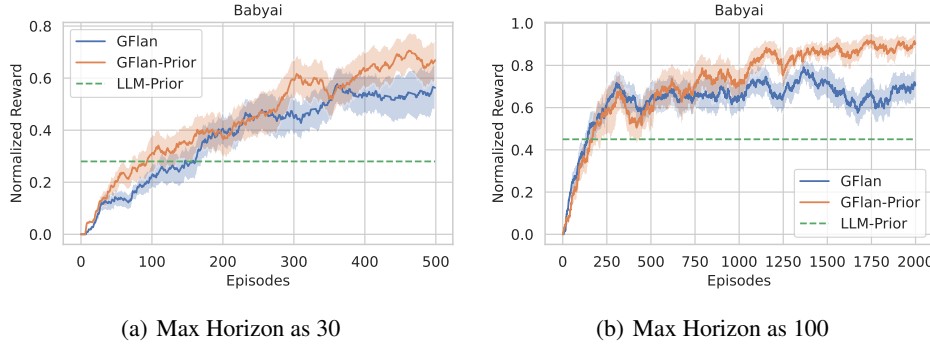

(a) Max Horizon as 30                    (b) Max Horizon as 100

Figure 7: The comparison of policy-based algorithms on BabyAI.

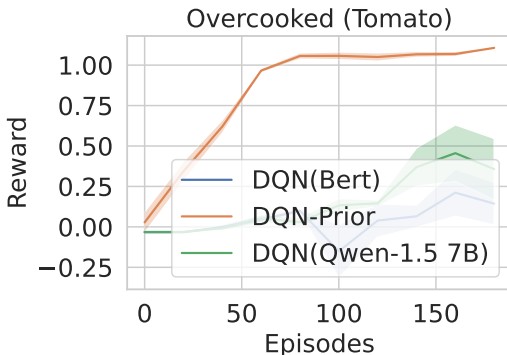

Figure 8: An ablation to verify the effectiveness of using the LLM to provide prior actions versus to encode state-action pairs.

To reduce the overhead of querying the LLM for every state, we store its outputs: For DQN-Prior, we store action proposals. For DQN (Qwen-1.5 7B), we store (s, a) embeddings.

As shown in the Figure 8 and Table 13, DQN-Prior outperforms both DQN (Qwen-1.5 7B) and DQN (BERT). By leveraging the LLM's high-level reasoning, DQN-Prior narrows the exploration space to a reliable action sub-space, improving sample efficiency. While DQN (Qwen-1.5 7B) benefits from the 7B LLM's superior semantic understanding and outperforms DQN (BERT), (s,a) embeddings cannot be directly used for action selection. Instead, a mapping from embeddings to Q-values must be learned from scratch to correctly rank the entire action space, thereby increasing sample complexity.

In addition, DQN-Prior requires less time and CPU memory compared to DQN (Qwen-1.5 7B). The higher memory and time consumption of DQN (Qwen-1.5 7B) are primarily due to its large embedding size (4096 dimensions), which significantly increases the cost of training a mapping (MLP adaptor) from state-action embeddings to Q-values, as well as the memory required to store these embeddings.

Table 13: The training costs of DQN-based baselines: DQN-Prior, DQN(BERT), DQN(Qwen1.5-7B). Both DQN (BERT) and DQN (Qwen-1.5 7B) involve the use of the 7B LLM. To minimize the overhead of querying the LLM for every state, we pre-store its outputs: for DQN-Prior, we store action proposals, while for DQN (Qwen-1.5 7B), we store (s, a) embeddings.

| Baselines | Training Time(min) | GPU Usage | CPU usage | Embedding Dimension |
|---|---|---|---|---|
| DQN-Prior | 13.8 | 28340 MB | 18KB | 768 |
| DQN(BERT) | 16.2 | 17864MB | / | 768 |
| DQN(Qwen-1.5 7B) | 74.8 | 22662MB | 22724MB | 4096 |

**Applying Direct Posterior Sampling on the Top of PPO**   Inspired by the success of direct posterior sampling in value-based reinforcement learning (RL), which effectively explores and updates the value function within the LLM prior action space, we recognize that LLMs can provide a narrow and reliable sub-action space. Building on this success, we aim to investigate policy-based RL within this action space. Specifically, we seek to learn a policy that approximates the soft Q-distribution over the LLM prior space:

$$\pi_\theta(a|s) \approx \frac{Q(s,a)}{\sum_{a' \in \mathcal{C}^k(s)} Q(s,a')}$$

In practice, we directly apply Proximal Policy Optimization (PPO) within the prior sub-action space. Formally, for the $i$-th iteration, the policy $\pi_{\theta_i}$ is learned by optimizing:

$$\arg\max_\theta \mathbb{E}_{\mathcal{C}^k(s_t) \sim p_{LLM}(\cdot|s_t),(s_t,a_t) \sim \pi_{\theta_{i-1}}(\mathcal{C}^k(s_t))} \text{CLIP}\left( \frac{\pi_\theta(a_t|s_t)}{\pi_{\theta_{i-1}}(a_t|s_t)} \right) \hat{A}_t + \eta\mathcal{H}(\pi_\theta(\cdot|s_t)) \quad (26)$$

where $A_t$ is the estimated advantage, $\eta$ is the hyperparameter that encourages exploration. In the textual environments we considered, we used GFlan Carta et al. (2023) to train PPO with Flan-T5 small as the action policy. For simplicity, we denote the policy-based algorithm as GFlan-Prior(Sub-Action Space) or GFlan-Prior(Sub AS).

The comparison of policy-based RL with LLM prior is shown in Figure 9. GFlan-Prior(KL) indicates the PPO algorithm with the incorporation of LLM prior through the addition of a KL-penalty. These results demonstrate that incorporating LLM prior into PPO training, either by adding a KL-penalty or by providing a sub-optimal sub-action space, can improve the sample efficiency of RL training. Additionally, while both GFlan-Prior(KL) and GFlan-Prior(Sub AS) are able to successfully complete the final task with rewards approaching 1, GFlan-Prior(Sub AS) achieves higher rewards due to its explicit exploration and optimization within the sub-action space, which is more efficient than the implicit use of LLM prior by adding a KL penalty.

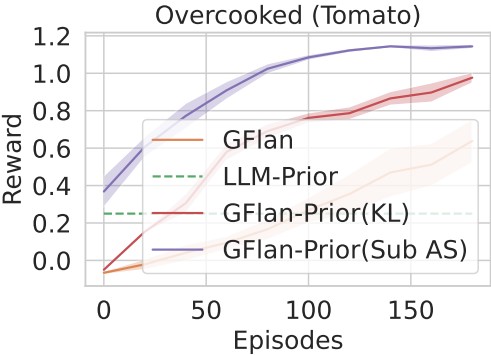

Figure 9: The comparison of policy-based algorithms with LLM prior.

## 10   ADDITIONAL RELATED WORKS

**Value-based Inference of LLM**   Not limited to the pre-training phase, it has been recently investigated how scaling law applies at inference time for LLM. This is often referred to as searching over an enormously large output space with guidance from value estimation. For example, the naive case is beam search (citation) which selects the top K possible sequence based on cumulative probabilities as opposed to greedy search. The value estimate here is the likelihood of the sentence as predicted by the language model. (Wang et al., 2022) propose CoT-SC, which searches over reasoning paths and selects the most frequent answer. Tree-of-Thought (ToT, (Yao et al., 2023)) adopts depth/breadth-first search. (Hao et al.) introduce Reasoning-via-Planning (RAP) which uses Monte-Carlo Tree Search and value estimate from prompting LLM. TS-LLM designed by (?) guides MCTS with a learned value function conditioned on state and a learned Outcome-supervised Reward Model (ORM). (Li et al., 2024) proposes Q-probing to adapt a pre-trained language model to also

maximize a tasks-specific reward function in code generation tasks. Furthermore, recent works investigate Process-supervised Reward Model (PRM, (Lightman et al., 2023)) and apply it in LLM inference, (McAleese et al., 2024) searches over a step-wise critic model in code reviewing tasks. (Wang et al., 2024a) learn a PRM in math reasoning and use it for LLM fine-tuning. (Snell et al., 2024) provides a detailed benchmark of different inference methods. (Zhang et al., 2024a) represents the score as the probability of a single text token (e.g. 'Yes' or 'No') under the context and the prompt. Even though our proposed method can be seen as an example of value-based inference, we focus on a different perspective where we investigate how LLM helps conventional value-based reinforcement learning algorithms in complex environments.

**LLM-based Reward Function for RL**  Beyond serving as high-level planners for SDM tasks, LLMs are also utilized for generating reward signals, which are subsequently used for RL training. For example, (Kwon et al., 2023) leverages LLMs to directly generate reward signals prompted with historical interaction information. Motif (Klissarov et al., 2023) employs LLMs to annotate pairwise preference datasets, which are then used to train a reward function. Given the challenges LLMs face in directly generating low-level actions for robotics tasks with continuous representations, several studies (Yu et al., 2023; Ma et al., 2024) have instead utilized the coding capabilities of GPT-4 to generate human-level reward code. These reward codes are then employed to facilitate RL training.

More discussion can be found in Section 2.

