# OpenReview forum: "Efficient Reinforcement Learning with Large Language Model Priors"
_ICLR.cc/2025/Conference — ICLR 2025 Poster_

### Official Review · Reviewer_uGBu · 2024-10-31

**Soundness:** 3
**Presentation:** 3
**Contribution:** 3
**Rating:** 6
**Confidence:** 4

**Summary:**

This paper proposes an approach to leverage LLM priors for RL, both in the online and offline settings.
The setting is restricted to text-based observations. Specifically, they estimate a prior p(a|s) by sampling K actions from the LLM conditioned on the state, and use these to either restrict the Q-values to the actions with non-zero probability (for online or offline Q-learning based methods), or to enforce a KL penalty with respect to the policy’s distribution over actions. They evaluate on several text-based environments and show improvements in terms of sample efficiency over vanilla RL. Overall, the paper does a decent job of illustrating the benefits of the method within its scope, but the scope itself is quite limited (in terms of broad applicability and scalability - more on this below). Also, there are a number of alternate methods for incorporating LLM priors into RL systems, that are not discussed or compared to.

=====
Update after rebuttal: the authors have added several additional experiments which partially address the initial limitations I discussed. Although I still think the environments are on the simple side, and the method in its current form is difficult to scale to high-throughput settings, the results are convincing enough in demonstrating the approach's merits and give hints of wider applicability to meet the bar for acceptance.

**Strengths:**

- the method is simple and makes intuitive sense. Its benefits are clearly demonstrated, although in fairly simple text-based environments. Although other works have proposed ways to incorporate LLM priors into RL systems, to my knowledge this particular approach is new.
- the paper is overall clearly written and easy to understand. I have a few suggestions below, but they are relatively minor.

**Weaknesses:**

- the method is limited to settings where the observation consists of text, which is a big limitation. One could imagine workarounds to apply this to images, where one first uses a pretrained captioning model to extract textual captions and then feed these to the LLM. However, it’s not clear if this would give enough granularity to produce a reasonable action prior and this would need to be checked experimentally.
- the method, as I understand it, will have difficulty scaling to high-throughput RL settings because it requires a call to the LLM for each observation encountered by the agent (which is computationally expensive). This also limits its applicability to small tasks or offline settings where the LLM outputs can be precomputed in parallel and cached. Are there any ways to address this? For example, could you distill the LLM action prior into a lightweight model and use this instead?
- there are other ways to incorporate LLM priors into RL systems, namely through the reward function rather than the action distribution. See for example the references below [1, 2, 3, 4] (this is not a complete list). It is currently unclear what the pros/cons of the proposed method are compared to this other family of approaches. It would be helpful to include at least one method from this family as a comparison in the experiments. Also, these are not discussed in the related work section.


[1] Eureka: https://arxiv.org/abs/2310.12931
[2] Language2Reward: https://arxiv.org/abs/2306.08647
[3] Motif: https://arxiv.org/abs/2310.00166
[4] Reward Design with LLMs: https://arxiv.org/abs/2303.00001

**Questions:**

If the authors can address any number of the points raised in the Weaknesses section, I would be willing to raise my score accordingly. I also have a few minor comments below:

- In Section 4.1, it would be helpful to give some more detail on the environments you use. The current descriptions are quite high level and not everyone may be familiar with these environments or have intuitions about them. Maybe include a figure with example observations/actions for each one - you still have half a page under the page limit so there should be space.
- Some of the references are missing dates, for example (Snell et al), line 232.
- This is a bit of a nitpick, but in Figure 1, is there a reason the OpenAI logo is on the LLM box? My understanding is that this paper does not use OpenAI’s LLMs, and instead uses open models (Qwen and Llama). I don’t think an institutional logo is necessary, but if you do want to include one, it would seem more fair to give credit to those that made the models that you use available.

---

> ### Author Response · Authors · 2024-11-19
> **Response to reviewer uGBu**
>
> We thank the reviewer for their time and effort in reviewing our paper, we are glad that you appreciate 1) the intuitive and simple method, 2) the novelty of our approach and 3) the clear writing of our paper. We have provided detailed explanations and clarifications to resolve your concerns and made several updates that have improved the paper based on your recommendation. We respectfully hope you can reconsider your score to reflect our updates.
>
> > Q1 The method is limited to the setting of text observation. Can it be applied to image environments
>
> Our primary goal is to explore how incorporating LLMs' prior knowledge can enhance RL training in sequential decision-making tasks. A number of **standard RL benchmarks** with discrete action spaces, such as StarCraft[1], Overcooked[2], Card games[3], and BabyAI[4], **feature symbolic observation or grid-based image observation that can be converted into text using rule-based methods**. Previous studies [1,2,3,4] have verified the feasibility of LLMs on solve these games. These textualized RL benchmarks are also suitable for our approach. In this work, we have demonstrated the effectiveness of our approach on several prominent decision-making tasks, including ALFWorld, Overcooked, and Frozen Lake.
>
> In addition, Atari-GPT [5] has demonstrated that multi-modal LLMs, such as GPT-4 and GPT-4V, can achieve sub-optimal performance on Atari tasks with image obervations. Building on their attempts, our framework could be readily adapted to image-based observation settings by **utilizing multi-modal LLMs to generate action priors**. We believe that leveraging our framework to further enhance the decision-making capabilities of multi-modal LLMs would be a worthwhile direction for future work.
>
> [1] Ma, Weiyu, et al. "Large language models play starcraft ii: Benchmarks and a chain of summarization approach." arXiv preprint arXiv:2312.11865 (2023).
>
> [2] Zhang, Ceyao, et al. "ProAgent: building proactive cooperative agents with large language models." Proceedings of the AAAI Conference on Artificial Intelligence. Vol. 38. No. 16. 2024.
>
> [3] Costarelli, Anthony, et al. "Gamebench: Evaluating strategic reasoning abilities of llm agents." arXiv preprint arXiv:2406.06613 (2024).
>
> [4] Carta, Thomas, et al. "Grounding large language models in interactive environments with online reinforcement learning." International Conference on Machine Learning. PMLR, 2023.
>
> [5] Waytowich, Nicholas R., et al. "Atari-GPT: Investigating the Capabilities of Multimodal Large Language Models as Low-Level Policies for Atari Games." arXiv preprint arXiv:2408.15950 (2024).
>
> > Q3: incorporate LLM priors into RL systems through the reward function rather than the action distribution.
>
> Thank you for your recommendations. Our work leverages the reasoning and high-level planning capabilities of LLMs to facilitate RL training by generating sub-optimal action priors, while the studies you recommended utilize LLMs to generate intrinsic reward functions through their powerful coding or in-context learning capabilities. Both our approach and the studies on LLM-based reward functions avoid the costly fine-tuning of LLMs. We have added a discussion of the recommended works to the related work section in the Appendix on page 23. Additionally, we have conducted experiments comparing our method with the LLM-based reward function approach [6] you recommended. Please refer to the response box below for details.
>
> [6] Kwon, Minae, et al. "Reward Design with Language Models." The Eleventh International Conference on Learning Representations. 2023
>
>
> >Q4: Give some more detail on the environments
>
> Thank you for your suggestion. We have included detailed reward shaping descriptions in Section 4.1 and added examples of observation and action representations for all environments in Appendix 9.1 on page 18.
>
> > Q5: About references missing dates and Fig 1.
>
> Thank you for carefully reviewing our paper and pointing out these issues. We have updated Fig. 1 and the references according to your suggestions.

---

> ### Author Response · Authors · 2024-11-19
> **Response to reviewer uGBu**
>
> > Q2: About querying LLM for each observation encountered by the agent (which is computationally expensive).
>
> Our work **leverages LLMs' domain knowledge to enhance RL sample efficiency**, crucial for real-world applications where online sampling is costly. Our methods significantly outperform baselines in both online and offline settings.
>
> Most studies of LLMs as agents for decision-making tasks [1, 2, 3, 4, 5] also face the challenge of querying the LLM for each observation encountered by the agent. In particular, RL fine-tuning approaches [3, 4, 5], which use large LLMs (up to 7B parameters) as the action policy, require querying the LLM at each step to collect data for online training. Reflect_RL [6] distills GPT-4's prior knowledge into a smaller LLM (GPT-2xl) but still relies on extensive querying of GPT-4 to generate a sufficiently comprehensive dataset for each specific task. Since the LLM prior is fixed, we can resort to the **caching technique** within our framework to **alleviate the need for querying the LLM at every step**, as per your suggestion.
>
> We introduce a variant of the DQN-Prior algorithm, called DQN-Prior (Cache), which uses cached action candidates for previously seen observations and queries the LLM prior, storing its outputs only for new observations. We present the time and computational costs of our DQN-Prior(query LLM each time) and DQN-Prior (Cache), as well as the RLFT baseline, TWOSOME [4], for learning to converge on Overcooked (Tomato) in the table below. Detailed results are illustrated in Appendix 9.4 on page 21. **The time-consuming of our DQN-Prior(Cache) is significantly lower than the TWOSOME**, while maintaining comparable performance.
>
> The caching technique is well-suited to our framework, as the 7B LLM is capable of providing a reliable sub-action space. DQN-Prior (Cache) significantly reduces query time costs but also achieves superior performance compared to DQN-Prior (querying the LLM at each step), as frequent queries introduce more uncertainty.
>
>
>
>
>
> |                                | Reward | Episodes | Training Time(min) | GPU usage | **Query LLM(Qwen-1.5 7B) times** |
> | ------------------------------ | ------ | -------- | ------------------ | --------- | -------------------------------- |
> | DQN-Prior(query LLM each step) | 0.97   | 180      | 31.1               | 28298MB   | 2933                             |
> | DQN-Prior(Cache)               | 1.11   | 180      | **13.8**               | 28340MB   | 434                              |
> | TWOSOME                        | 1.16   | 640      | 60.4               | 25014MB   | 5000                             |
>
> [1] Yao, Shunyu, et al. "ReAct: Synergizing Reasoning and Acting in Language Models." The Eleventh International Conference on Learning Representations. (2023)
>
> [2] Shinn, Noah, et al. "Reflexion: Language Agents with Verbal Reinforcement Learning." Advances in Neural Information Processing Systems 36 (2023).
>
> [3]Wen, Muning, et al. "Reinforcing LLM Agents via Policy Optimization with Action Decomposition." The Thirty-eighth Annual Conference on Neural Information Processing Systems.2024
>
> [4]Tan, Weihao, et al. "True Knowledge Comes from Practice: Aligning Large Language Models with Embodied Environments via Reinforcement Learning." The Twelfth International Conference on Learning Representations.2024
>
> [5]Christianos, Filippos, et al. "Pangu-agent: A fine-tunable generalist agent with structured reasoning." arXiv preprint arXiv:2312.14878 (2023). 2023
>
> [6] Zhou, Runlong, Simon S. Du, and Beibin Li. "Reflect-RL: Two-Player Online RL Fine-Tuning for LMs." arXiv preprint arXiv:2402.12621 (2024).

---

> ### Author Response · Authors · 2024-11-22
> **Additional results on comparison with language model to generate reward**
>
> Dear reviewer,
>
> We would like to add an additional experiment comparing our approach with using LLMs to generate rewards to further address your concerns. Our work leverages an LLM to generate sub-optimal prior actions for each state, thereby reducing the large action space into a manageable one, thus eliminating the need to explore meaningless actions and improving sample efficiency. Following your suggestion, we include a comparison with [1], which uses LLMs to generate reward signals using in-context examples for RL training.
>
> The ALFWorld environment **lacks dense reward signals**, providing only a reward of 1 for successfully completing the task, and **features large action spaces** with a maximum of up to 50 actions, making it challenging to solve. We implement the baseline [1] of using an LLM to generate auxiliary rewards on ALFWorld, called DQN-LLM2R (DQN with LLM-generated Rewards) for simplicity. The comparisons of baselines are shown in the table below. Specifically, we use **LLM (Qwen-1.5 7B for a fair comparison) to generate an auxiliary  reward for each state**, and the prompt we used is:
>
> ```
> Task Description: {instruction}
> Current Observation: {state_text}
> Based on the current observation and its alignment with task completion, please select an appropriate reward.
> For example:
> 1. The task is 'Place the spray bottle on the toilet.' Rewards are: 0.2 for finding the spray bottle, 0.4 for picking it up and holding it, 0.6 for moving toward the toilet with it, 0.8 for reaching the toilet with it, and 1.0 for placing it on the toilet.
> 2. The task is 'Put a candle on the toilet.' Rewards are: 0.2 for locating the candle, 0.4 for picking it up and holding it, 0.6 for moving toward the toilet with it, 0.8 for reaching the toilet with it, and 1.0 for placing it on the toilet.
> Choose a reward from these options: [0.0, 0.2, 0.4, 0.6, 0.8, 1.0].
> Reward:
> ```
> |           | Win Rate |
> | --------- | -------- |
> | DQN-Prior | 0.8      |
> | DQN       | 0        |
> | DQN-LLM2R | 0        |
> |LLM-Prior(Qwen-1.5) 7B|0.2|
>
> The complete failure of DQN-LLM2R may stem from two primary factors: 1. It's quite likely that the **7B LLMs assign incorrect reward signals, leading the Q-function to learn in the wrong direction**, and 2. **the large action space, which makes it hard for DQN to explore subgoals**. However, our DQN-Prior can effectively solve the task by leveraging the high-level action reasoning and planning capabilities of LLMs to simplify the exploration challenge. In addition, while the LLM may be suboptimal in determining the correct action, achieving a win rate of only 0.2 when directly taking actions, our algorithm performs RL within a suboptimal sub-action space, effectively leveraging the prior knowledge of LLMs and improving error tolerance on action generation from LLM.
>
> Examples of LLM generated reward are given as:
>
> * Example 1:
>
> *Task: put a soapbottle in toilet. Observation: You arrive at loc 13. On the countertop 1, you see a candle 2, a sink 1, a soapbar 1, a toiletpaper 3, and a toiletpaper 1.*
>
> *LLM output ['1.0\n\nGreat task! Here are some']*  **Incorrect**
>
> * Example 2:
>
> *Task: put a candle in toilet. Observation : You pick up the candle 1 from the countertop 1.*
>
> *LLM output ['0.2\nExplanation: The task is to']* **Correct**
>
> * Example 3:
>
> *Task: put a handtowel in garbagecan. Observation: You arrive at loc 18. The drawer 1 is closed.*
>
> *LLM output [ '0.8 for reaching the toilet with the hand']* **Incorrect**
>
> * Example 4:
>
> *Task: put a candle in toilet. Observation: You arrive at loc 10. On the garbagecan 1, you see a spraybottle 1, and a toiletpaper 2..*
>
> LLM output *['0.2\nExplanation: The task is to']* **Incorrect**
>
> [1] Kwon, Minae, et al. "Reward Design with Language Models." *The Eleventh International Conference on Learning Representations*. 2023
>
> We sincerely appreciate the valuable comments on improving our paper. We hope that our responses sufficiently address your concerns. We are eager to participate in ongoing discussions throughout the reviewer-author discussion period if you have further comments or concerns. Please let us know if you have any further questions or concerns and we are very happy to address them.

---

> ### Author Response · Authors · 2024-11-25
> **Additional experiments on applying our method to image-based environment**
>
> Dear reviewer uGBu,
>
> We sincerely thank you for your valuable comments on improving our work. As per your suggestion, we have added an experiment to validate the feasibility of our method in image-based environments by integrating our algorithm with a Vision-language LLM.
>
> We conducted experiments on Frozen Lake, a grid-based game with image observations. In this setup, we used **Qwen2-VL-7B** to generate action candidates based on the image observations and then perform RL within this prior subspace.
>
> As shown in the table below, our DQN-Prior (Qwen2-VL-7B) achieves superior performance compared to both the standard DQN (when trained with the same number of episodes) and Qwen2-VL-7B. This demonstrates that by performing exploration within the prior action space provided by Qwen2-VL-7B, our method significantly enhances the sample efficiency of DQN and delivers a performance boost over Qwen2-VL-7B.
>
>
> | **Method**              | **Performance** (Winning rate) |
> | ----------------------- | ------------------------------ |
> | DQN                     | 0.55                           |
> | DQN-Prior (Qwen2-VL-7B) | 0.93                           |
> | Qwen2-VL-7B|0.10|

---

> > ### Author Response · Authors · 2024-11-27
> > **Looking forward to further discussion**
> >
> > Dear reviewer uGBu,
> >
> > We sincerely thank you for your valuable comments on improving our work. We would like to respectfully highlight the updates made in response to your suggestions and to address your concerns:
> >
> > 1) We have verified that our method is plug-and-play, adapting seamlessly to image-based environments by utilizing Vision-Language LLMs to process image observations and generate prior actions.
> > 2) We have demonstrated that our approach can leverage caching techniques to reduce the need to query the LLM at every time step while maintaining high performance.
> > 3) We have added a comparison between using the LLM as a suboptimal action prior and using it for reward generation.
> >
> >
> > We respectfully hope that you will take these updates and our responses into account when making the final decision. As the author-reviewer discussion period is coming to an end, we look forward to further engaging with you. We would be happy to continue the discussion and provide additional information if you have any further concerns or questions.

---

> > > ### Comment · Reviewer_uGBu · 2024-11-27
> > >
> > > Thank you for the thorough response, and for running several of the requested experiments. My concerns have been partially addressed:
> > >
> > > - the DQN-cache method improves the computational efficiency somewhat, by about 2x. Although this will probably still not enable applications to very high-throughput settings, it is a step in the right direction. A potential idea to explore in the future could be to distill the LLM action predictions in a small parametric model (possibly in an online manner). Specifically, you can learn a distribution over actions conditioned on the state with a small model, using the LLM action predictions as targets. This could then be used to apply your method in very high throughput settings, replacing the LLM by the learned model over time.
> > >
> > > - the addition of the new baseline is helpful in adding context to the proposed method, thank you for adding this.
> > >
> > > - the addition of the VLM experiments, although still on a simple environment, provide some evidence that this method could apply more broadly beyond purely text-based observations, which is a good (although still preliminary) sign.
> > >
> > > Based on these improvements, as well as some of the added paper clarifications, I am raising my score by two grades, to a 6.

---

> > > > ### Author Response · Authors · 2024-11-28
> > > >
> > > > We sincerely thank you for your careful review and valuable suggestions to improve our work. We are pleased that you recognize 1) the integration of caching techniques into our framework as 'a step in the right direction' in reducing high computational overhead, 2) the additional experiments on language-to-reward, and 3) VLM applications provide evidence of broader applicability beyond purely text-based observations. The reviewer raises an interesting point, and we think that distilling the LLM action prior into a smaller model is a promising approach to further enhancing computational efficiency in future work.

---

### Official Review · Reviewer_8Lkv · 2024-11-02

**Soundness:** 3
**Presentation:** 2
**Contribution:** 1
**Rating:** 6
**Confidence:** 4

**Summary:**

Authors present a new way to train LLM-based agents in interactive sequential environments. Their approach is based on leveraging a potentially large frozen LLM to generate action subsets for encountered states. The action subset is then used to simplify the RL finetuning of a smaller LLM. Their approach can be adapted to value-based (giving a subset of actions for the LLM Q-network) and policy-based scenarios (constraining the LLM policy with a KL-div loss over the subset of actions). Authors showcase how their approach outperforms baselines in TextWorld and Overcooked, both in online and offline learning settings.

**Strengths:**

This paper tackles a very important topic: how to leverage LLMs into sequential decision-making tasks incorporating language.
I enjoyed reading it.
Performance improvements over considered baselines are impressive.

**Weaknesses:**

### Value-based version

I am having trouble understanding how useful the value-based formulation is. What I mean is that, from my understanding, at inference time you need both the frozen LLM and the trained Q-network (which is a smaller LLM). This makes the approach quite expensive regarding memory. Regarding inference, it is also costly: the frozen LLM must provide k outputs before being able to use the Q-network LLM, for each step in the environment.
A good comparative study here to motivate this approach could be to have a baseline which discards the frozen-LLM. What happens in this scenario ? Is it what you call the DQN baseline ? Here, it would be interesting to see what happens if you use a larger LLM during training (something which induces a memory burden comparable to using a frozen LLM + a smaller trained Q-net LLM).

### Comparative study with GFlan from carta et al

The only baseline considered from the litterature is GFlan from carta et al. While I am not expert enough to assess whether this is sufficient, I have concerns about this comparative study.
In carta et al, GFlan is evaluated on a text-based version of the BabyAI gridworld environment:

Why not featuring this environment in the study ?

Any intuitions on why GFlan is failing completely in ALFWorld (TextWorld) while it does maintain some performances in Overcooked ?






### Clarity

While the paper is mostly clear, I had trouble understanding the exact training and inference pipeline. The paper could benefit an update of figure 1 (along with providing the policy-based version of figure 1 somewhere).The right side of Figure 1 could be improved: in the current version, it is hard to visually understand that the Q function is used to weight each input actions and that these weights are used for proportional sampling of the next action. Also in figure 1 it is hard to understand that there are two pre-trained LLMs being used, one frozen generating an action subset, and a smaller one trained and used as a critic. I only understood that there was two pretrained LLMs being used in the DQN-Prior baseline at line 395 "... by combining the Q-function, trained with Qwen-1.5 7B". I would suggest making it clear early on.

l.97-99 "we first generate a free-form output from the LLM, for example, a 7B LLM, which is then mapped to an executable action through a simple rule-based projection."
--> I would recommend moving at least part of what is discussed in the appendix into the main body of the paper, to be clearer about this.
Related to this: I only understood that you repeat this output-to-action mapping multiple times at line 166-167. It would be clearer to mention it sooner in the paper.


### Minor / Typos

There are 3 acronym definitions in the abstract, which is a lot. Consider reducing this number, e.g. dropping SDM which is used only once.

l31. "human-AI dialogue (McTear, 2022; Li et al., 2019)," --> make sure to sort citations in ascending order wrt year. This error appears multiple times.

l. 232 --> (Snell et al.) missing year in citation.

l.104 "by (Levine, 2018)" --> use \cite not \citep when directly mentioning the citation in the sentence.

l.304 "can be found in the Appendix" -->  refer to the appropriate appendix section(s) rather than a general reference

l.323 "while avoiding full" --> falling

l.352 "Behavior Clone(BC)." --> Behavioral Cloning ?

Results-related figures could be moved closer to the result analysis in section 4.3 to simplify reading

l.404 "k = 1 indicates the performance of the LLM on its own, without involving the Q-function"
--> Why calling this scenario k=1 and not k=0 ? you could theoretically use your frozen LLM to create a single action subset.

**Questions:**

l.286 "We plot the rewards averaged over the final third of the training processes for trainable baselines" --> Evaluation of a model performance is not performed by conducting periodic performance measurements over a test set of tasks ? Could you elaborate on how performance is measured ?

In your policy-based version, which you present in section 3.3 as an extension of the work of Carta et al., 2023, I am not sure to understand what is trained is what is fixed. Are you, like in Carta et al, training all parameters of the LLM ? What is being trained by your modified PPO loss ? Would be good to have the policy-based version of figure 1 somewhere. My understanding after reading section 4.2 is that you have a large fixed pretrained model used to compute action priors, and you train another smaller Flan-T5 as the policy.

In your offline learning experiments, by looking at table 1 it looks like you are applying a vanilla DQN to offline datasets. It comes with the risk of overestimating values of actions that are not supported by the dataset. Would it make more sense to use value-based methods designed for offline learning, e.g. Implicit Q Learning from Kostrikov et al ?

In your offline learning results, I see that CQL underperforms wrt Behavioral Cloning. This is unusual: any ideas on why ?




Given the aforementioned questions and potential limitations of this paper, I will go for a weak reject, and look forward to the discussion to update my score.

---

> ### Author Response · Authors · 2024-11-19
> **Response to reviewer 8Lkv**
>
> We thank the reviewer for their time and effort in reviewing our paper, we are glad that you appreciate the important topic and impressive performance improvements of our work. We have provided detailed explanations and clarifications to resolve your concerns and made several updates that have improved the paper based on your recommendation. We respectfully hope you can reconsider your score to reflect our updates.
>
> > Q1: Value-based version: (1) requires both the frozen LLM and the trained Q-network, which is quite memory-intensive; (2) includes a baseline that discards the frozen LLM.
>
> In our paper, we set the Q-network to learn an adapter (a 3-layer MLP) on top of the BERT embedding, with the BERT model (which has only 110M parameters) fixed. As a result, due to **the small size of the BERT model and the fact that only the adapter (a 3-layer MLP) is being learned**, the computational memory and cost of the Q-network remain manageable.  We have clarified the setting of Q-network in lines 224-229 on page 5.
>
> Our DQN-Prior explores and updates within the prior action space, sampling $k$ suboptimal action candidates from the frozen 7B LLM. Indeed, the DQN baseline discards the frozen LLM and explores the entire action space. As shown in Fig. 3, our DQN-Prior significantly improves sample efficiency compared to DQN, thanks to the manageable yet reliable prior action space, especially in ALFWorld, where DQN fails to learn due to the large action space and sparse reward signal.
>
> > Q2: Comparison on BabyAI Gridworld Environment
>
> Thank you for your suggestion. We will include a comparison on the BabyAI environment in the reversion.
>
> > Q3: Any intuitions on why GFlan is failing completely in ALFWorld (TextWorld) while it maintains some performance in Overcooked?
>
> There are two main reasons for this phenomenon:  1) ALFWorld has a large action space, with up to 50 possible actions, compared to Overcooked, which has a maximum of only 8 possible actions.  2) The Overcooked environment provides dense reward shaping signals, while ALFWorld offers sparse reward signals—1 for successfully completing a task and 0 otherwise. Please refer to section 4.1 on page 6 for detailed reward settings.
>
> > Q4: About Clarity and typos
>
> We sincerely thank you for your valuable suggestions. We have updated the draft point by point according to your suggestions.
>
> Specifically, about the question of "k = 1 indicates the performance of the LLM on its own, without involving the Q-function" --> Why calling this scenario k=1 and not k=0 ? you could theoretically use your frozen LLM to create a single action subset."
>
> In this scenario, the frozen LLM generates a single action, resulting in an action subset containing only that action, thus we call this scenario as k=1.
>
> > Q5: About the clarity of the policy-based version:
>
> Yes, you are correct. We use the frozen Qwen-1.5 7B (or 14B for the more complex Overcooked Salad task) LLM to generate prior actions and train a Flan-T5 small LLM (with fewer than 1B parameters) as the action policy. **We have added an illustration of the policy-based RL with LLM prior, as shown in Fig. 2 on page 5.**
>
> > Q6:  Value-based methods designed for offline learning makes more sense than DQN due to the overestimation risk.
>
> [1] has verified that off-policy DQN can be applied to offline, discrete action space settings but tends to underperform compared to its variants that use multiple Q-estimations to mitigate overestimation. The CQL we tested, a value-based offline RL algorithm, is specifically designed to address the overestimation of out-of-distribution state-action pairs. This work introduces CQL-Prior and validates that optimizing within the prior action space significantly enhances the sample efficiency compared to CQL.
>
> [1] Agarwal, Rishabh, Dale Schuurmans, and Mohammad Norouzi. "An optimistic perspective on offline reinforcement learning." International conference on machine learning. PMLR, 2020.

---

> ### Author Response · Authors · 2024-11-19
> **Response to reviewer 8Lkv**
>
> > Q7: Why does CQL underperform Behavioral Cloning?
>
> Thank you for your question. The main reason for this result may be the **insufficient data volume**, which limits CQL's ability to effectively discriminate the optimal path. Since the offline dataset contains many successful trajectories, detailed in Table 11 of the appendix, BC can learn from these successful trajectories and attain moderate performance with fewer examples. Additionally, we add a group of results on the offline setting of Overcooked(Salad) in Table 1, which are also briefly illustrated below. These results indicate that **CQL is limited by the data volume**, BC is constrained by the ratio of good to bad data. In contrast, our **CQL-Prior demonstrates greater robustness to both sample volume and data distribution**.
>
> |   state-action pairs           | CQL  | DQN  | BC   | CQL-Prior |
> | ------------ | ---- | ---- | ---- | --------- |
> | Salad(1000)  | 0.32 | 0.14 | 0.57 | **1.01**      |
> | Salad(24000) | **1.33** | 0.32 | 0.78 | **1.31**     |
>
> > Q8:  "We plot the rewards averaged over the final third of the training processes for trainable baselines" --> Evaluation of a model performance is not performed by conducting periodic performance measurements over a test set of tasks ? Could you elaborate on how performance is measured ?
>
> Yes, you are correct. The performance of online training is assessed through periodic evaluations of the latest model. The statement "We plot the rewards averaged over the final third of the training processes for trainable baselines" is incorrect and should be removed. Really thank you for pointing that out.

---

> ### Author Response · Authors · 2024-11-22
> **Additional results on BabyAI**
>
> Dear reviewer 8Lkv,
>
> We have added additional experiments on BabyAI as you suggested. We compare our policy-based algorithm, GFlan-Prior, with GFlan in the BabyAI environment. We note that this task is challenging due to the partially observed nature of the environment and the random initialization of the map for each episode. We focus on the BabyAI task 'Go to the red ball', setting the maximum horizon at 30 and training both GFlan and GFlan-Prior with 500 episodes. The comparison of baselines in BabyAI is presented in the table below, and a detailed results illustration can be found in Appendix 9.4 Fig.7 on page 21. Our GFlan-Prior, which integrates both LLM prior knowledge and environmental interaction experience, also accommodates the partially observed BabyAI and outperforms both GFlan and the LLM-prior.
>
>
>
> |             | Reward |
> | ----------- | ------ |
> | LLM prior   | 0.28   |
> | GFlan       | 0.55   |
> | GFlan-prior | **0.68**  |
>
> We sincerely appreciate the valuable comments on improving our paper. We hope that our responses sufficiently address your concerns. If you have further comments or concerns, we are eager to participate in ongoing discussions throughout the reviewer-author discussion period. Please let us know if you have any further questions or concerns and we are very happy to address them.

---

> > ### Comment · Reviewer_8Lkv · 2024-11-23
> > **Answer**
> >
> > I am pleased to see that authors answered many of my concerns, and will raise my score to 6, and therefore **recommend acceptance**
> >
> > To improve my score again I would need:
> >
> > * more experiments and/or details regarding the BabyAI comparison. In figure 5 of Carta et al paper I see that their agent reaches 0.95+ success rate on the "GoTo" task. Is this GoTo task ==  'Go to the red ball' ? If yes, then why are your reproduced performances for GFlan lower than this ?
> >
> > * "A good comparative study here to motivate the DQN-based approach could be to have a baseline which discards the frozen-LLM used to compute action priors, and just use something bigger than BERT . I feel like this experiment could bring value, as (if your solution is better) it justifies to allocate memory for a frozen LLM computing action priors VS just "going bigger" for the 1B BERT

---

> ### Author Response · Authors · 2024-11-23
>
> Dear Reviewer,
>
> We are delighted that our response addressed many of your concerns and that you recommend acceptance. We would greatly appreciate it if you could kindly update the score in the openreview comment box.
>
> We thank you for your further detailed review and questions, our responses are as follows:
>
> 1. **Experiment Details on BabyAI**:
>   Yes, we conducted experiments on the task "Go to the red ball" based on the codebase provided by Carta et al GFlan.  The main performance differences from Carta et al paper may result from the following factors:
>
>    - The original BabyAI environment sets the maximum horizon to 100, but we reduced it to 30 to avoid situations where random navigation could achieve the goal and to more effectively evaluate the algorithm's decision-making ability.
>    - In addition, the original GFlan paper (Figure 5) reported results after 1.5 million training steps using Flan-T5 large (780M parameters). In contrast, our results were obtained with fewer than 15K training steps (using 1% of the training examples) and Flan-T5 small (80M parameters), for the limited training time and computational resource considerations. Furthermore, our GFlan-prior also performed better than GFlan under the same training settings (including the same horizon, training steps, and action policy LLM).
>
>    Moreover, we ran GFlan on the task "Go to the red ball" with a maximum horizon of 100, fewer than 150K training steps, and Flan-T5 small as the action policy. Under these settings, GFlan achieved a winning rate of 0.75, and our GFlan-Prior achieved a winning rate of 0.9. The winning rate of LLM-prior is 0.45.
>
> 2. **Compare to a DQN-Based Approach with Larger LLM for Q-Value Estimation**:
>    Thank you for your valuable suggestion. We agree that this comparison could verify whether leveraging prior knowledge from LLMs for action priors is more effective than improving state-action embeddings by using a larger model than BERT. We plan to include a comparison using larger LLMs (such as Qwen-1.5 7B for a fair comparison) for the Q-network to explore this idea.
>
> Best Regards,

---

> ### Author Response · Authors · 2024-11-24
> **Additional Experiment on using a Large LLM for the Q-network**
>
> Dear reviewer,
>
> We really thank you for your valuable comments on improving our work. We would like to include an additional experiment to compare the effectiveness of using a large LLM for generating action priors versus improving state-action embeddings.
>
> We compared three baselines on Overcooked(Tomato):
>
> 1. **DQN-Prior**: Uses Qwen-1.5 7B to generate a sub-action space and BERT to encode (s, a) pairs for Q-value estimation.
> 2. **DQN (BERT)**: Uses BERT to encode (s, a) pairs and performs DQN across the full action space.
> 3. **DQN (Qwen-1.5 7B)**: Uses Qwen-1.5 7B to encode (s, a) pairs and performs DQN across the full action space.
>
> **To reduce the overhead of querying the 7B LLM for every state, we store its outputs**: For **DQN-Prior**, we store action proposals. For **DQN (Qwen-1.5 7B)**, we store (s, a) embeddings.
>
> As shown in the table below and Fig 8 in Appendix on page 22, **DQN-Prior outperforms both DQN (Qwen-1.5 7B) and DQN (BERT)**. By leveraging the LLM’s high-level reasoning and planning ability, **DQN-Prior** **narrows the exploration space to a reliable action sub-space, improving sample efficiency**. While DQN (Qwen-1.5 7B) benefits from the 7B LLM’s superior semantic understanding and outperforms DQN (BERT), The (s,a) embeddings from the 7B LLM also cannot be directly used for action selection. Instead, a mapping from embeddings to Q-values must be learned from scratch to **correctly rank the whole action space**, thereby increasing sample complexity.
>
> In addition, **DQN-Prior requires less time and CPU memory compared to DQN (Qwen-1.5 7B)**. The higher memory and time consumption of **DQN (Qwen-1.5 7B)** are primarily due to its large embedding size (4096 dimensions), which significantly increases the cost of training a mapping (MLP adaptor) from state-action embeddings to Q-values, as well as the memory required to store these embeddings.
>
> |                   | Reward | Training Time | GPU Memory | CPU Memory for storing LLM's output | Embedding Dimension|
> | ----------------- | ------ | ------------- | ---------- | ----------------------------------- | ---------------------------- |
> | DQN-Prior         | **1.11**   | **13.8**      | 28340MB    | **18KB**                                | 768                          |
> | DQN(Bert Q value) | 0.21   | 16.2          | 17864MB    | /                                   | 768                          |
> | DQN(Qwen1.5-7B)   | 0.46   | 74.8          | 22662MB    | 22724MB                             | 4096                         |
>
> We hope that our responses have solved your concerns and respectfully hope you can consider the final decision accordingly. As the author-reviewer discussion period is coming to an end, we are respectfully looking forward to further discussing with you. Please let us know if you have any additional questions or concerns, we would be more than happy to address them.
>
> Best regards,
>
> The authors

---

> > ### Comment · Reviewer_8Lkv · 2024-11-25
> > **answer**
> >
> > I thank reviewers for providing these additional details. I am convinced by the additional experiments with a larger LLM.
> > Comparative study with carta et al work is good, but could have been better (not only on the simpler setting, for longer training time, with Flan-T5 large).
> >
> > As such, I will keep my score.
> >
> > > Moreover, we ran GFlan on the task "Go to the red ball" with a maximum horizon of 100, fewer than 150K training steps, and Flan-T5 small as the action policy. Under these settings, GFlan achieved a winning rate of 0.75, and our GFlan-Prior achieved a winning rate of 0.9. The winning rate of LLM-prior is 0.45.
> >
> > This is the most interesting comparative analysis, as it is closest to their setup (max horizon of 100). How many seeds did you use ? It would be good to have learning curves for this experiment.

---

> ### Author Response · Authors · 2024-11-26
> **About experiments on BabyAI**
>
> Dear reviewer,
>
> We are glad that you recognize the additional experiments with a larger LLM, and that “the comparative study with Carta et al.'s work is good.”
>
> We use three random seeds for training and present the training curve in Figure 7(b) in appendix on page 21. Due to limited rebuttal time, it was challenging to fine-tune the 780M Flan-T5 Large with a vast number of training steps, so we opted for simpler setups.  Despite this, our method consistently outperforms GFlan on the same training episodes. These results already offer compelling evidence for our main claim that incorporating prior knowledge of LLMs enhances the sample efficiency of RL training.
>
> Best regards

---

### Official Review · Reviewer_vRpg · 2024-11-03

**Soundness:** 4
**Presentation:** 4
**Contribution:** 3
**Rating:** 8
**Confidence:** 4

**Summary:**

In this work, the authors propose idea of using large language models (LLMs) for providing prior information about execution of a task, which simplifies exploration in reinforcement learning (RL) and improves sample efficiency. The core idea is to treat LLMs as prior distribution over actions and then apply variational inference in conjunction with posterior sampling. The authors demonstrate this technique on both value-based and policy-based approaches. In value-based implementation, a subset of the action space is chosen for Q-learning instead of using entire action space. This subset selection is done using LLM prior. Similarly, in the policy-based implementation, a KL-penalty is introduced with respect to the prior LLM action distribution, but implemented differently from regular KL penalty in RL from human feedback (RLHF), by posterior sampling of multiple candidate actions. The experiments are conducted on text-based RL environments with large action spaces and multi-step horizon (note this is crucial for demonstrating the difference against RLHF techniques). Value-based baselines include DQN for online learning and CQL for offline learning. For policy-based baseline, PPO algorithm is compared against.

The results demonstrate the superiority of the approach in (1) overall returns, (2) generalizing beyond seen tasks, (3) sample efficiency.

**Strengths:**

The paper is written very clearly with appropriate figures, well-formatted equations and more importantly lucid logical flow. The contributions of the paper are specified along with particular sections corresponding to individual contribution. Preliminaries provide enough details and are not overexplained.

The core idea of the paper is pretty neat -- LLMs are trained on large amount of data and they have knowledge about tasks if not fine-grained controllability; this knowledge is useful to guide an RL training.

I like how the value-based implementation is designed -- sampling a set of actions using LLM prior, taking Q estimates, sampling actions using their Q estimates. The method is well grounded in the theory where direct posterior sampling is shown to be proportional to LLM prior multiplied with soft-Q distribution. The idea being simple, can be plugged with both online and offline Q-learning approaches.

The versatility of the idea is shown by applying it to policy-based RL. The LLM prior guided learning takes form similar to KL-regularized RL, hence modifying PPO in a straightforward way would lead to easy access to prior knowledge. However, I do I have some contentions around the novelty of this part which is discussed in weakness section.


The choice of the experimental setup is also very apt. The environments are chosen so that LLMs can provide textual action descriptions which then can be executed through text to action conversion in the environment's processing.

**Weaknesses:**

Practicality of value based approaches: Although the value-based direct posterior sampling technique described in the paper is logically sound, I wonder about its practicality. Mainly, value-based approaches when applied to combinatorically large action spaces like text generation is practically infeasible. Hence, the paper's proposed method might be hard to apply beyond toy textual environments. Noticeably, these environments are designed to accept actions in a particular format or use rigorous action decoding, etc. to deal with the combinatoric complexity. Such environments are rare in reality.

Problems with choice of prior inducing LLMs: Two LLMs might produce semantically same action but different text. I am curious to know whether the Q-learning based approaches would assign similar values to both. In general, I feel that prior inducing LLM and its compatibility with the environment's action decoding plays major role when they are used to train RL policies in text-based sequential decision making.

Novelty of the additional KL penalty in policy-based RL: ELBO-based RL approaches [1, 2] have been already established in the RL literature. The additional KL-penalty proposed in the current paper is based on the same. In single step decision making, the KL-penalty on an LLM prior would be exactly same as KL-penalty used along with supervised fine-tuned (SFT) policy. Thus, in that setting, the paper's technique lacks novelty. Now, coming to the multi-step decision making, unless the proposed direct posterior sampling is also applied on top of PPO, additional KL-penalty is equivalent to using a good enough prior policy (e.g. behavior cloned policy). Therefore, the novelty of the paper is limited in policy-based RL.

Combined together,  I find that paper provides great insights into a direct posterior sampling paradigm for using prior knowledge present in the LLMs, but the approach might have limited applicability beyond hand-designed textual environments, its applicability is limited by limitations of value-based approaches, and its novelty in policy-based RL is not significant.

**Questions:**

Questions and Suggestions:

- Should the proportionality sign on line 147 be equal to? I understand the sentence is correct in its current form.

- The $p$ is overloaded on lines 112-113, especially, please change the "$p_{LLM}$ as the action prior $p$" sentence where the latter $p$ is said to be not confused with $p$ used for probability of start state or action given a state in the definition of probability of trajectory.

- I am not sure by what is meant by "Denote that the above …" in Proposition 1. Please rewrite this for clarification.

- Line 215 - "reply buffer" should be changed to "replay buffer"

- Is the gamma in equation 8 the same as the discount factor?

- Line 323 - "avoiding full into holes". Please rectify this.

---

> ### Author Response · Authors · 2024-11-19
> **Response to Reviewer vRpg**
>
> We thank the reviewer for their time and effort in reviewing our paper, we appreciate your recognition of 1) the clear writing and logical flow, 2) the neat idea of using LLMs' knowledge to guide RL training, and 3) the appropriate experimental setups of our work. We have provided detailed explanations and clarifications to resolve your concerns and made several updates that have improved the paper based on your recommendation. We respectfully hope you can reconsider your score to reflect our updates.
>
> > Q1  Practicality of value based approaches: ."....Noticeably, these environments are designed to accept actions in a particular format or use rigorous action decoding, etc. to deal with the combinatoric complexity. Such environments are rare in reality."
>
> Our primary goal is to explore how incorporating LLMs' prior knowledge can enhance RL training in textual sequential decision-making tasks. Several **textual decision-making benchmarks** such as ALFWorld, Science World, and Webshop have been proposed to evaluate LLMs as agents. Additionally, many **standard RL benchmarks** with discrete action spaces, such as StarCraft[1], Overcooked[2], and Card games[3], **feature symbolic observation and action representations that can be converted into text using rule-based methods**. These text-based environments have well-structured action spaces, making them suitable for our value-based approach. In this work, we demonstrate the effectiveness of our value-based approach on several prominent decision-making tasks, including ALFWorld, Overcooked, and Frozen Lake.
>
> [1] Ma, Weiyu, et al. "Large language models play starcraft ii: Benchmarks and a chain of summarization approach." arXiv preprint arXiv:2312.11865 (2023).
>
> [2] Zhang, Ceyao, et al. "ProAgent: building proactive cooperative agents with large language models." Proceedings of the AAAI Conference on Artificial Intelligence. Vol. 38. No. 16. 2024.
>
> [3] Costarelli, Anthony, et al. "Gamebench: Evaluating strategic reasoning abilities of llm agents." arXiv preprint arXiv:2406.06613 (2024).
>
> > Q2 " Problems with choice of prior inducing LLMs: Two LLMs might produce semantically same action but different text. I am curious to know whether the Q-learning based approaches would assign similar values to both. "
>
> Thank you for your question. During implementation, we prompt the LLMs with the textual state and admissible actions. The **LLMs then generate a free-form text output, which is mapped to a textual executable action using a simple rule-based method**, i.e., directly extracting the action appearing in the LLM output. Through this process, the Q-network processes only the textual executable action, ignoring other irrelevant information from the LLMs. Please refer to lines 102–107 for more details and lines 972-987 on page 19 in the appendix for an example of mapping LLM outputs to actions.
>
> > Q3 Novelty of the additional KL penalty in policy-based RL: Compared to ELBO-based RL approaches and supervised fine-tuned (SFT) policy in single-step decision-making tasks
>
> From a variational inference perspective, our algorithm falls under the category of ELBO-based RL approaches. This work presents an **effective pathway to integrate LLMs' rich domain knowledge into policy-based RL by utilizing it as an informative action prior** within the traditional ELBO-based RL framework.
>
> The objective of our policy-based RL is similar to the formulation in RLHF settings, where a language model is fine-tuned with KL regularization to preserve its original language capabilities. In our work, LLMs are used for providing suboptimal action priors for challenging sequential decision-making tasks, while the actual belief (posterior) is derived by training a smaller language model to distill both task-relevant LLM priors and environmental experience.
>
>
> > Q4 "Should the proportionality sign on line 147 be equal to? I understand the sentence is correct in its current form"
>
> We use the proportionality sign because we omit the term $p(\mathcal{O}=1|s_t)$, which is irrelevant to action and not always equal to 1.
>
> >Q5 "The is overloaded on lines 112-113, especially, please change the " as the action prior " sentence ......"
>
> Thank you for your suggestion. We have revised the sentence for clarity: "Here, we specify $p_{LLM}(a|s)$  as the action prior $p(a|s)$." In our work, $p$ denotes any general probability distribution, while  $p_{LLM}(a|s)$  is specified as a concrete distribution for $p(a|s)$  as the action prior within this framework.
>
> We sincerely thank you for carefully reviewing our paper and providing valuable suggestions. We have revised the draft point-by-point following your review.

---

> > ### Comment · Reviewer_vRpg · 2024-11-21
> >
> > Thanks for the prompt answers to my concerns.
> >
> > I understand the text-based RL environments have concept of admissible actions and when the LLM is conditioned to output one of them, they would be restricted to a finite action space and hence value-based approach would work. In real world too, if we were to hand-design the action space and allow for semantically similar text actions to be generated and map them to admissible action space, the value-based approach would work; I can see this after reading your answer.
> >
> > Next, I see where the proposed method fits among the ELBO-based RL and its similarities to policy-gradient RLHF. The proposed method brings a new perspective to the standard RLHF; however, in terms of the novelty of the objective there is only incremental contribution, I feel.
> >
> > I also appreciate authors changing the paper to accommodate the suggestions.
> >
> > In all, I feel that this work shows how LLMs trained on vast knowledge can provide a guidance mechanism to avoid unnecessary exploration in RL environments and learn policies efficiently. I find the foundational idea sound, the theory very supportive, the experiments indicative of the idea's prowess, and feel this work would be helpful for future research along use of pretrained agents for helping learn decision-making in new related environments. I am increasing my score to 8.

---

### Official Review · Reviewer_h6X1 · 2024-11-04

**Soundness:** 2
**Presentation:** 3
**Contribution:** 2
**Rating:** 5
**Confidence:** 4

**Summary:**

This paper considers treating LLMs as prior action distributions and integrating them into RL frameworks. Specifically, authors proposes to use LLM for exploration,  Q target calculation and imposing conservatism. Experiments on the ALFWorld and Overcooked environments illustrates the effectiveness of the proposed framework.

**Strengths:**

1. This paper is clearly written and easy to follow.
2. The considered "incorporating the background knowledge of LLM into RL" is an interesting topic.

**Weaknesses:**

1. There are no convergence guarantees of sampling using LLM as prior for Exploration and Q-function update. Moreover, there is no guarantee that learning the Q-function as shown in Equation (7) will give us a conservative Q-function.
2. If the LLM can not provide a good prior, constraining the learned policy as shown in Equation (8) will result in sub-optimal policies.
3. The RL process has to query the LLM each time we update the policy, which is time-consuming and computation-consuming.

**Questions:**

1. The experiments are only conducted on simple environments with discrete action spaces. What if continuous action space (e.g., offline RL on the D4RL benchmark)?

---

> ### Author Response · Authors · 2024-11-19
> **Response to Reviewer h6X1**
>
> We thank the reviewer for their time and effort in reviewing our paper, we are glad that you appreciate the clarity and the interesting topic of our work. We have provided detailed explanations and clarifications to resolve your concerns and made several updates that have improved the paper based on your recommendation. We respectfully hope you can reconsider your score to reflect our updates.
>
> > Q1: About the convergence guarantees of sampling using LLM as prior for Exploration and Q-function update
>
> Thank you for your question. This work presents an effective approach to incorporating LLM prior knowledge to enhance value-based RL by exploring and optimizing within the prior action space. Proposition 1 on page 4 shows that **our posterior sampling strategy over the prior action subspace approximates soft sampling over the entire action space**, guided by the LLM prior distribution and Q-values. The reviewer makes an interesting point. We think convergence guarantees for Q-learning and CQL within the prior space are worth investigating in future theoretical analysis.
>
> Empirical results in Fig. 3 and Table 1 demonstrate that DQN-Prior and CQL-Prior based on the LLM prior space significantly outperform baselines and improve sample efficiency. Specifically, we present a set of offline results on Overcooked (Salad) in the table below, where CQL-Prior significantly reduces data requirements compared to CQL for successfully completing a task, while achieving comparable performance when sufficient data is available. These results **empirically demonstrate that optimization within the prior action space barely hinders CQL convergence while improving sample efficiency**.
> | State-action pairs | CQL| BC   | CQL-Prior |
> | -- | -- | --| --|
> | Salad(1000)  | 0.32     | 0.57 | **1.01** |
> | Salad(24000) | **1.33** | 0.78 | **1.31**  |
>
>
> > Q2: "If the LLM can not provide a good prior, constraining the learned policy as shown in Equation (8) will result in sub-optimal policies"
>
> Thank you for your question. We agree that a poor LLM may not provide helpful information, and the incorporation of the LLM prior, as shown in Eq. (8), may not directly assist in learning the optimal policy. The results in Fig. 4(a) and (c) also demonstrate this phenomenon: the LLM prior itself may be suboptimal for decision-making tasks, and a large coefficient for the KL constraint (between the learned action policy and the LLM prior) negatively impacts the performance of the learned action policy. In fact, LLMs only provide priors, the actual belief (posterior) about the scenario is derived from our training within the environment. This approach, to some extent, alleviates the issue of biased priors. As shown in Fig. 4(a), although the LLM prior itself may be suboptimal for decision-making tasks, the variational posterior approximation in Eq. (8) successfully learns a proficient action policy. In addition, we can reduce the reliance on the LLM prior by using a smaller KL coefficient, or by implementing a parameter decay mechanism.

---

> ### Author Response · Authors · 2024-11-19
> **Response to Reviewer h6X1**
>
> >Q3: About time-consuming and computation-consuming for querying the LLM each time.
>
> Our work leverages LLMs' domain knowledge to **enhance RL sample efficiency**, crucial for real-world applications where online sampling is costly. Our methods significantly outperform baselines in both online and offline settings.
>
> Previous RL fine-tuning approaches [1, 2, 3], which use large LLMs (up to 7B parameters) as the action policy, also require querying the LLM at each step to collect data for online training. Since the LLM prior is fixed, we can resort to **the caching technique** within our framework to **alleviate the need for querying the LLM at every step**.
>
>  We introduce a variant of the DQN-Prior algorithm, called DQN-Prior (Cache), in which we use cached action candidates for seen states and query the LLM prior, storing its outputs only for unseen states. We present the time and computational costs of our DQN-Prior(query LLM each step) and DQN-Prior (Cache), as well as the RLFT baseline, TWOSOME [2], for learning to converge on Overcooked (Tomato) in the table below. Detailed results are illustrated in Appendix 9.4 on page 20-21. **The time-consuming of our DQN-Prior(Cache) is significantly lower than the TWOSOME**, while maintaining comparable performance.
>
> The caching technique is well-suited to our framework, as the 7B LLM is capable of providing a reliable sub-action space. DQN-Prior (Cache) significantly reduces query time costs but also achieves superior performance compared to DQN-Prior (query LLM each step), as frequent queries introduce more uncertainty.
>
> |       | Reward | Episodes | Training Time(min) | GPU usage | **Query LLM(Qwen-1.5 7B) times** |
> | -------- | ------ | -------- | ------------------ | --------- | ------- |
> | DQN-Prior(query LLM each step) | 0.97   | 180      | 31.1  | 28298MB   | 2933   |
> | DQN-Prior(Cache)  | 1.11   | 180      | **13.8**               | 28340MB   | 434  |
> | TWOSOME | 1.16   | 640      | 60.4               | 25014MB   | 5000    |
>
> [1]Wen, Muning, et al. "Reinforcing LLM Agents via Policy Optimization with Action Decomposition." The Thirty-eighth Annual Conference on Neural Information Processing Systems.2024
>
> [2]Tan, Weihao, et al. "True Knowledge Comes from Practice: Aligning Large Language Models with Embodied Environments via Reinforcement Learning." The Twelfth International Conference on Learning Representations.2024
>
> [3]Christianos, Filippos, et al. "Pangu-agent: A fine-tunable generalist agent with structured reasoning." arXiv preprint arXiv:2312.14878 (2023). 2023
>
> >Q4: About continuous action space
>
> Thank you for your question. As you mentioned, this work focuses on investigating the incorporation of LLM background knowledge into RL by letting the LLM generate reliable action candidates. However, **it is challenging for LLMs to directly generate continuous actions [6]**. Previous attempts to use LLMs for solving continuous RL tasks have also avoided outputting continuous actions. For instance, [4] uses LLMs to **plan over pre-defined high-level skills** (e.g., "pick up an object"), which then form a discrete action space, similar to the setting in our paper. Other studies use LLMs to **generate code for enhancing state representation [5] or designing reward functions [6]**. Our work leverages the LLM's high-level action reasoning and planning abilities; therefore, this work focuses on discrete action spaces. It’s important to note that the action space size in ALFWorld is not as small as imagined, with up to 50 possible actions, making exploration still challenging.
>
> [4] Lin, Kevin, et al. "Text2motion: From natural language instructions to feasible plans." Autonomous Robots 47.8 (2023): 1345-1365.
>
> [5] Wang, Boyuan, et al. "LLM-Empowered State Representation for Reinforcement Learning." ICML (2024).
>
> [6] Yu, Wenhao, et al. "Language to Rewards for Robotic Skill Synthesis." Conference on Robot Learning. PMLR, 2023.

---

> ### Author Response · Authors · 2024-11-22
> **Looking forward to further discussion**
>
> Dear Reviewer h6X1,
>
> We sincerely appreciate the valuable comments on improving our paper. We hope that our responses sufficiently address your concerns. If you have further comments or concerns, we are eager to participate in ongoing discussions throughout the reviewer-author discussion period. Please let us know if you have any further questions or concerns and we are very happy to address them.

---

> > ### Comment · Reviewer_h6X1 · 2024-11-25
> > **Thanks to your responses**
> >
> > Thanks to your detailed responses. However, some of my concerns remain.
> >
> > > Proposition 1 on page 4 shows that our posterior sampling strategy over the prior action subspace approximates soft sampling over the entire action space, guided by the LLM prior distribution and Q-values.
> >
> > Proposition 1 requires that $k \to \infty $. However, in your experiments (as shown in Table 2), the maximum $k$ you choose is 15.
> >
> > > However, it is challenging for LLMs to directly generate continuous actions.
> >
> > I understand that it may be not trivial for continuous action space. However, I believe that whether from a research perspective or a practical perspective, investigating continuous action spaces is essential for current RL studies. And there are some emerging works, like [1].
> >
> > I keep my scores.
> >
> >
> >
> > [1] Pang, J. C., Yang, S. H., Li, K., Zhang, J., Chen, X. H., Tang, N., & Yu, Y. KALM: Knowledgeable Agents by Offline Reinforcement Learning from Large Language Model Rollouts. In *The Thirty-eighth Annual Conference on Neural Information Processing Systems*.

---

> ### Author Response · Authors · 2024-11-25
> **Further Response to Reviewer h6X1**
>
> Dear Reviewer,
>
> We really thank you for your valuable comments on improving our work. We are glad that our response has addressed your concerns regarding computational cost and the potential for sub-optimal policy results due to a poor LLM prior. We will now address your concerns about action sample number k and continuous actions.
>
> > 1. Proposition 1 requires that $k \to \infty$. However, in your experiments (as shown in Table 2), the maximum \( k \) you choose is 15.
>
> Thank you for your question. In Proposition 1, in the limit of $k \to \infty$, the sampled actions can exactly approximate the LLM prior action distribution, $p_{\text{llm}}(a|s)$, by the law of large numbers. Empirically, we sample k actions to approximate the LLM prior distribution and set $k=5$ for most training scenarios to balance the computational cost and approximation accuracy. Even with a small $k = 5$, our method outperforms other baselines, as shown in Figures 3 and 4 and Table 1, demonstrating that k = 5 provides a sufficient approximation for LLM prior distribution. This is further demonstrated in Figure 4(b) (page 9) and Figure 5 in the appendix (page 20), where $k = 5$ already significantly outperforms $k = 1$.
>
> >2. I understand that handling continuous action spaces may not be trivial. However, I believe that investigating continuous action spaces is essential for current RL research, both from a theoretical and practical perspective. There are also some emerging works, such as [1].
>
> Thank you for your recommendation. Our main contribution lies in investigating how to leverage the rich prior knowledge of LLMs to enhance the sample efficiency of RL training. We achieve this by developing an efficient approach that uses **the fixed LLM as a suboptimal action prior**. The work you referenced, [1], also highlights that “LLMs are inherently text-based and may be incompatible with numerical environmental data.” Thus, this work **fine-tunes the LLM** to adapt it to continuous action space environments.
>
> Enabling text-based LLMs to directly output continuous actions is beyond the scope of this work's contribution. Instead, we prioritize discrete action spaces, where most general-purpose LLMs are already capable of functioning without fine-tuning, to effectively validate our idea that rich knowledge in LLMs can assist in RL training. We thank the reviewer's recommendation, and we plan to discuss related works on LLMs for continuous spaces in the related work section.
>
> [1] Pang, J. C., Yang, S. H., Li, K., Zhang, J., Chen, X. H., Tang, N., & Yu, Y. KALM: Knowledgeable Agents by Offline Reinforcement Learning from Large Language Model Rollouts. In *The Thirty-eighth Annual Conference on Neural Information Processing Systems*.
>
> We hope that our responses have solved your concerns and respectfully hope you can reconsider your final decision accordingly. As the author-reviewer discussion period is coming to an end, we are respectfully looking forward to further discussing with you. Please let us know if you have any further questions or concerns and we are very happy to address them.
>
> Best regards,

---

> > ### Author Response · Authors · 2024-11-29
> > **Kindly looking forward to further discussions**
> >
> > Dear Reviewer h6X1,
> >
> > As the author-reviewer discussion period is coming to a close, we would like to kindly inquire whether our responses have addressed your concerns. Your participation in this process has been invaluable, and we greatly appreciate your time and thoughtful consideration. We look forward to further discussions, and we are more than happy to address any additional questions or concerns you may have. Thank you in advance for your continued support.
> >
> > Best regards,

---

> > > ### Comment · Reviewer_h6X1 · 2024-11-29
> > > **Official Comment by Reviewer h6X1**
> > >
> > > Thanks to your responses. I decide to maintain my score because the weaknesses and limitations I raised were not fully addressed during the rebuttal process.

---

### Meta-Review · Area_Chair_npQ2 · 2024-12-19

**Metareview:**

The authors propose to use LLM priors to guide policies in more efficient exploration. Specifically, they use Bayesian inference to integrate action distribution priors provided by an LLM and perform posterior sampling. They show how this can be done in both policy-based and value-based RL frameworks, and demonstrate improved sample efficiency in simple, discrete-action environments.

Reviewers found the paper easy to follow and clearly written. The core idea is simple and neat, and performance improvements over the chosen baselines are impressive. However, there were some reviewer concerns about the simplicity of the environments chosen for the experiments, and that the framework does not allow for continuous action environments. There was also no discussion in related work of how this framework compares to providing LLM priors via a reward function.

The authors provided additional experiments during the rebuttal phase that assuaged some reviewers' concerns. Given the nice idea, provision of both value-based and policy-based frameworks for incorporating prior action distributions, and alternative perspective from the more common reward-based framework, I vote to accept this paper.

**Additional Comments On Reviewer Discussion:**

There was some disagreement in the reviewer discussion about the paper. Reviewer h6X1 points out that lack of expertise by the LLM will lead to suboptimal policies, and that the proposed methods are limited to discrete action space. However, reviewer uGBu believes the results are convincing enough in demonstrating the approach's merits and give hints of wider applicability to meet the bar for acceptance, and reviewer vRpg agrees.

---

### Decision · Program_Chairs · 2025-01-22

Accept (Poster)